# Subsurface oxygen defects electronically interacting with active sites on $In_2O_3$ for enhanced photothermocatalytic $CO_2$ reduction

Weiqin Wei[1], Zhen Wei [2], Ruizhe Li[1], Zhenhua Li[3], Run Shi [3], Shuxin Ouyang [1✉], Yuhang Qi[4], David Lee Philips[2] & Hong Yuan[1]

Oxygen defects play an important role in many catalytic reactions. Increasing surface oxygen defects can be done through reduction treatment. However, excessive reduction blocks electron channels and deactivates the catalyst surface due to electron-trapped effects by subsurface oxygen defects. How to effectively extract electrons from subsurface oxygen defects which cannot directly interact with reactants is challenging and remains elusive. Here, we report a metallic In-embedded $In_2O_3$ nanoflake catalyst over which the turnover frequency of $CO_2$ reduction into CO increases by a factor of 866 (7615 $h^{-1}$) and 376 (2990 $h^{-1}$) at the same light intensity and reaction temperature, respectively, compared to $In_2O_3$. Under electron-delocalization effect of O-In-(O)$V_o$-In-In structural units at the interface, the electrons in the subsurface oxygen defects are extracted and gather at surface active sites. This improves the electronic coupling with $CO_2$ and stabilizes intermediate. The study opens up new insights for exquisite electronic manipulation of oxygen defects.

[1] Key Laboratory of Pesticide and Chemical Biology of Ministry of Education, College of Chemistry, Central China Normal University, Wuhan 430079, China. [2] Department of Chemistry, University of Hong Kong, Pokfulam Road, Hong Kong SAR, China. [3] Key Laboratory of Photochemical Conversion and Optoelectronic Materials, Technical Institute of Physics and Chemistry, Chinese Academy of Sciences, 100190 Beijing, China. [4] Chemical Engineering Institute, Hebei University of Technology, 300131 Tianjin, China. ✉email: oysx@mail.ccnu.edu.cn

Oxygen defects play a key role in the adsorption and activation of substrates and have attracted widespread attention in the field of catalysis[1–4]. They are made available for most of reactions involving photocatalysis, electrocatalysis, thermocatalysis, and photothermocatalysis. In recent years, applications of oxygen defects have made significant progress in $CO_2$ reduction[5–7], CO oxidation[8–10], and $NH_3$ synthesis[11–13]. But these previous studies focused on surface oxygen defects, especially increasing the density of active sites to enhance apparent catalytic activity, while the "quality" (namely, reactivity) of oxygen defects is usually neglected.

To date, the methods of creating oxygen defects include liquid-phase reduction, CO or $H_2$ reduction, thermal annealing in oxygen-deficient environment, flame reduction and electrochemical reduction, and interface engineering[4]. Among them, $H_2$ reduction treatment is relatively simple and does not introduce other undesired impurities. However, besides surface oxygen defects, subsurface or even deeper counterparts are also produced by this method[14,15]. The latter is distributed below the surface layer of 5–10 nm and cannot directly interact with substrates. Usually, every oxygen defect retains two electrons when neutral coordinated O atom is removed. One feasible redox reaction at oxygen defects must require effective electron exchange between oxygen defect and substrate to weaken chemical bond of the substrate molecule. Nonetheless, the electrons of oxygen defects are usually fettered by these oxygen defects (Coulomb interaction from adjacent metal ions)[8,16], retarding electron delivery to substrates and greatly reducing the catalyst activity.

Scientists have found that the metal loading on a semiconductor surface is beneficial to charge delocalization of the active sites on a semiconductor surface[16–19]. Intensively adopted metals were generally transition metals. Non-transition metal, such as In, has been reported to possess superior charge-conducting capability[20] and hence it could be considered to delocalize charges. Furthermore, $In_2O_3$ is an ideal catalyst for studying oxygen defects because of the richness and controllability of oxygen defects[7,21–25]. Therefore, we consider a special microstructure design of the catalyst to transfer such electrons bound in the subsurface oxygen defects to the surface oxygen defects via introduction of metallic In, which is expected to promote intrinsic activity. Inspired by this, $In_2O_3$ nanoflakes containing embedded metallic In were constructed, wherein In is a native element of $In_2O_3$, and thus it possesses better compatibility and affinity with an $In_2O_3$ lattice compared to foreign metal elements. The embedded metallic In is competent for constructing subsurface-surface electron channels, which reverses the disadvantage of charge localization by subsurface oxygen defects. Detailed characterizations and performance evaluations demonstrate that the configuration of metallic In embedded in $In_2O_3$ lattice can promote the electrons of subsurface oxygen defects to transport to surface oxygen defects, which improves the "quality" of the active sites and thereby boosts the intrinsic activity of $CO_2$ reduction (turnover frequency (TOF)).

## Results and discussion

**Temperature-dependent surface reconstruction of $In_2O_3$ nanoflakes**. Using indium nitrate and urea as raw materials, amorphous $In(OH)_3$ was prepared via a simple hydrothermal method, followed by calcination dehydration to obtain cubic bixbyite $In_2O_3$ (Supplementary Fig. 1). Then $In_2O_3$ was annealed in an atmosphere of mixed $H_2$ and Ar ($V_{H2}/V_{Ar} = 1/9$). As displayed in Fig. 1a and Supplementary Fig. 2, only surface reduction occurs before 300 °C whereas surface/subsurface simultaneous reduction to form metallic In appears at the temperature above 450 °C[14,26]. Accordingly, the catalyst (In-Em $In_2O_3$) comprising

metallic In was prepared. It is worth noting that the further reduction to In-Em $In_2O_3$ does not cause any surface reduction as shown in Supplementary Fig. 2 because the surface reduction degree of In-Em $In_2O_3$ has reached the maximum. X-ray diffraction (XRD) pattern (Supplementary Fig. 3) shows characteristic diffraction peaks of metallic In. In X-ray photoelectron spectra (XPS) (Supplementary Fig. 4), In and O elements are observed in $In_2O_3$ and In-Em $In_2O_3$ and the inclusion of other elements is excluded. Field emission scanning electron microscope (FE-SEM, Supplementary Fig. 5) demonstrates the two-dimensional irregular overall morphologies of $In_2O_3$ and In-Em $In_2O_3$. Transmission electron microscope (TEM, Fig. 1b) presents the nanoflake morphology of $In_2O_3$. In-Em $In_2O_3$ displays a compact profile with shrinking size and largely decreased specific surface area relative to $In_2O_3$ (Fig. 1c, Supplementary Fig. 6, and Supplementary Table 1) but with a similar strain effect (Supplementary Fig. 7). In the high-resolution TEM image (Fig. 1d), the lattice fringes with an interplanar spacing of 2.92 and 2.72 Å for $In_2O_3$ (222) and metallic In (101) facets can be observed, respectively. $In_2O_3$ phase of In-Em $In_2O_3$ preserves the pristine facets of $In_2O_3$ (Fig. 1d and Supplementary Fig. 8). Different from the visually dark oxides, the white color of the dots is due to the absence of lattice O atoms forming a lower-density stacking structure for easier TEM electron transmission.

The metallic In is most likely embedded into $In_2O_3$ nanoflakes according to the following three facts. First, based on in-situ reduction characteristics of $In_2O_3$[14], surface oxygen atoms are removed firstly whereupon subsurface counterparts are out to leave some voids on the surface and metallic In fills the voids (Fig. 1a). Second, numerous subsurface oxygen defects were generated and buried by metallic In (referring to temperature-programmed desorption ($CO_2$-TPD) of Fig. 2b). Third, atom force microscopy (AFM) images (Fig. 1e, f and Supplementary Fig. 9) exhibit a noticeable difference in the undulate surface height between the embedded (The highest is ~4 nm.) and supported structures (the highest is ~6 nm, and the height of In nanoparticle in AFM is ~2 nm (Supplementary Fig. 10).), excluding the possibility of a metal-supported structure. Unlike metal nanoparticles loaded on an oxide, the metal-embedded structure can firmly immobilize metallic In, which is similar to Ni nanocatalyst embedded in a hierarchical $Al_2O_3$ matrix[27] and prevent metal diffusion and agglomerate. Before the performances are discussed, it is worth mentioning that only surface oxygen defects interact with $CO_2$, whereas subsurface counterparts are not accessible to $CO_2$ (Fig. 1g). However, subsurface oxygen defects surrounding metallic In can indirectly interact with $CO_2$ (referring to the statements in the last two parts).

**Subsurface oxygen defects surrounding metallic In**. Surface compositions of the catalysts were tracked via XPS spectra. The peaks of the $In_{3d}$ core level at 444.7 and 452.3 eV are assigned to the characteristic spin-orbit splitting $3d_{5/2}$ and $3d_{3/2}$, respectively[25] (Supplementary Fig. 11). Compared with $In_2O_3$, the peaks of In-Em $In_2O_3$ shift to lower energy on account of the $In^0$ component[28]. Through peak deconvolution to In-Em $In_2O_3$, two groups of characteristic splitting peaks are attributed to $In^{3+}$ (444.7 and 452.3 eV, atom percent: 20.3%) and $In^0$ (444.1 and 451.7 eV, atom percent: 79.7%), respectively (Supplementary Fig. 11). The former belongs to $In_2O_3$ phase whereas the latter aggregates to form metallic In in the $In_2O_3$ lattice. On the other hand, $O_{1s}$ XPS peaks for the catalysts can be deconvoluted into three bands at 529.7, 531.3, and 532.5 eV (Fig. 2a), attributed to lattice oxygen, oxygen in the vicinity of the oxygen defects, and surface −OH, respectively[21,22]. Although $In_2O_3$ was not treated with $H_2$, the phase transformation from $In(OH)_3$ to $In_2O_3$ also

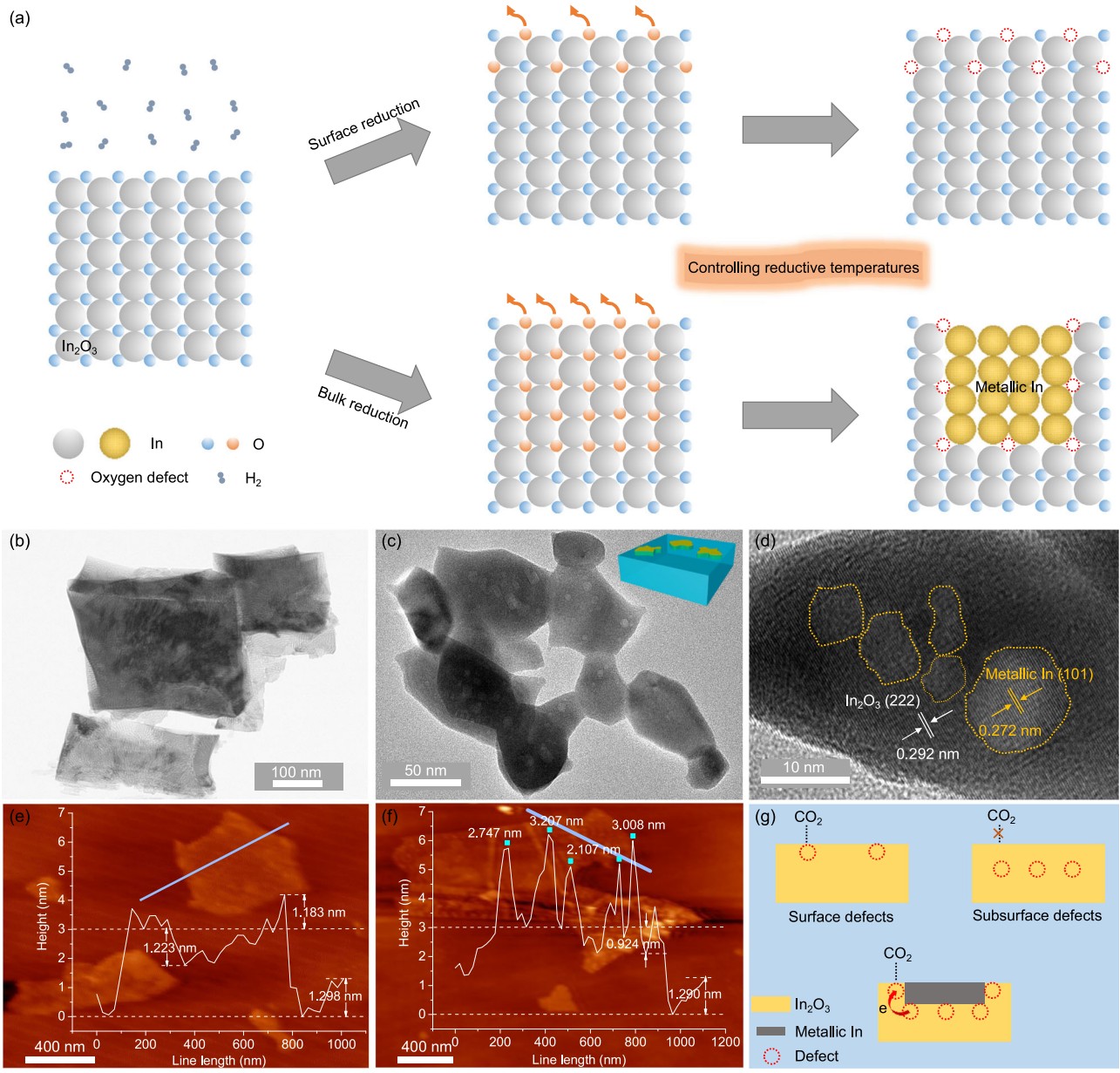

**Fig. 1 Structure/morphology. a** Schematic temperature-dependent surface reconstruction. TEM images of (**b**) $In_2O_3$ and (**c**) In-Em $In_2O_3$. **d** HRTEM image of In-Em $In_2O_3$. AFM images of (**e**) In-Em $In_2O_3$ and **f** $In_2O_3$ with metallic In nanoparticles loaded on the surface (Inset: the height along the blue line). **g** Scheme of three types of oxygen defects.

generated oxygen-vacancy defects[29], due to undercoordinated sites. $In_2O_3$ and In-Em $In_2O_3$ exhibit the same binding energies pertaining to oxygen defects, for electrons in oxygen defects are principally centered on three adjacent In atoms rather than O atoms[20,24,25]. The binding energy of lattice oxygen of In-Em $In_2O_3$ is lower than that of $In_2O_3$, implying a weakened binding of In and O in In-Em $In_2O_3$ and its electron richness[30]. The concentrations of the oxygen defects are measured to be virtually equivalent for the two catalysts (Supplementary Fig. 12 and Supplementary Table 2). But their color and light absorption properties are dramatically different because the light absorption characteristics of metallic In endows In-Em $In_2O_3$ full-spectrum absorption (referring to the last paragraph in the performance part).

CO$_2$-TPD profiles of the catalysts were measured to verify the number/type of oxygen defects on the surface. $In_2O_3$ presents marked chemisorption bands in CO$_2$-TPD pattern (Fig. 2b).

During catalysis at $T > 300\,°C$, the sites for chemisorption function as active sites of CO$_2$ reduction[7,26]. Unexpectedly, CO$_2$ uptake on In-Em $In_2O_3$ becomes almost negligible and thus oxygen defects of In-Em $In_2O_3$ seem not responsible for CO$_2$ chemisorption. Nevertheless, most of the reports have evidenced that oxygen defects of $In_2O_3$ are active sites of CO$_2$ reduction and the $O_{1s}$ XPS peak in Fig. 2a proves the existence of oxygen defects of In-Em $In_2O_3$. As is well-known, XPS technology can probe the components of several-nanometer depth[31] in line with the size of the metallic In of In-Em $In_2O_3$, whereas the TPD profile correlates with the adsorption property of a gas-solid interface. We speculate that oxygen defects of In-Em $In_2O_3$ exist at the interface, within a subsurface depth of 1–5 nm[31], thereby mitigating the CO$_2$ adsorption amount. Furthermore, the chemisorption appears by using HCl to remove a part of the metallic In, giving more powerful evidence on some subsurface oxygen defects being buried by metallic In. In comparison, with

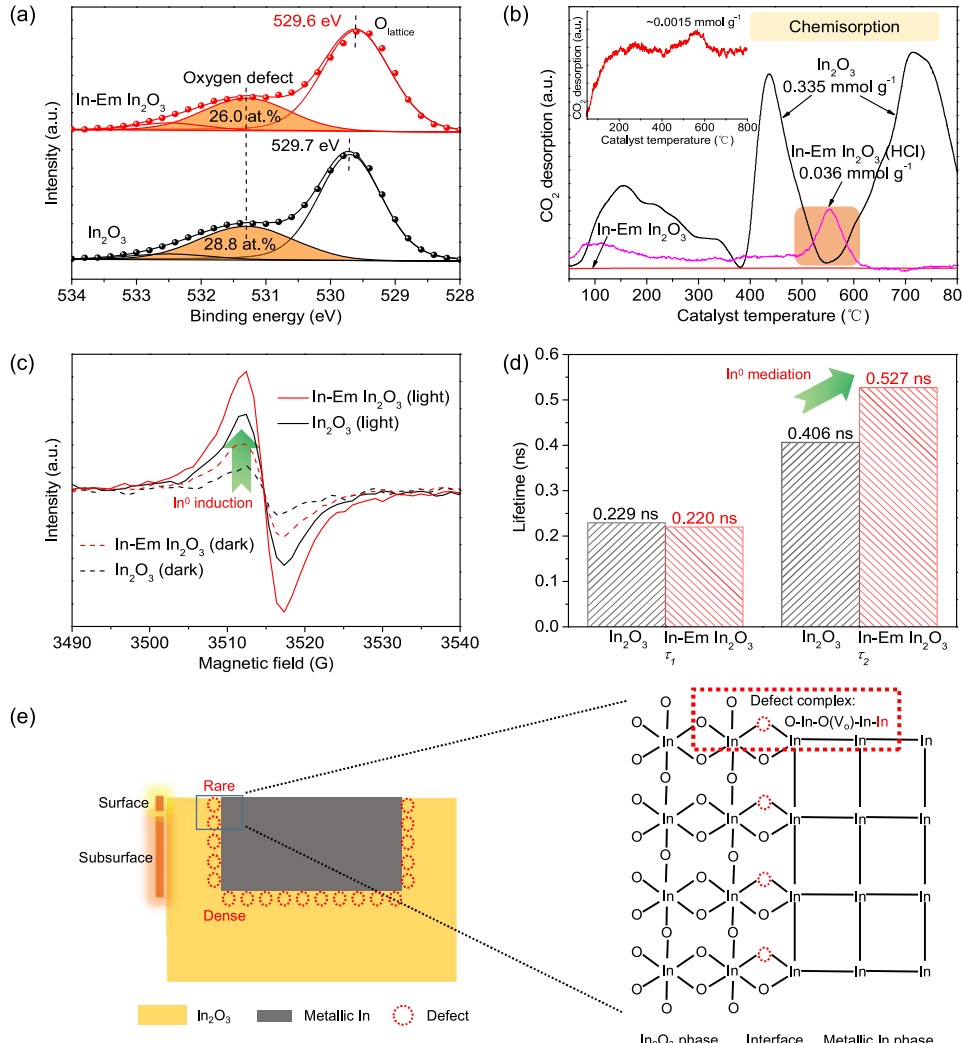

**Fig. 2 Identification of oxygen defects. a** $O_{1s}$ XPS spectra of $In_2O_3$ and In-Em $In_2O_3$. **b** $CO_2$-TPD profiles of $In_2O_3$, In-Em $In_2O_3$, and In-Em $In_2O_3$(HCl) with HCl etching. **c** ESR spectra of $In_2O_3$ and In-Em $In_2O_3$ at 100 K in the dark and under light irradiation. **d** Lifetimes ($\tau$) of positrons in PAS spectra of $In_2O_3$ and In-Em $In_2O_3$. **e** Model of distribution and structure of oxygen defects of In-Em $In_2O_3$.

$In_2O_3$ etched by HCl, the $CO_2$ adsorption amount does not change, excluding the effect of HCl on the $In_2O_3$ phase of In-Em $In_2O_3$ for increasing $CO_2$ adsorption (Supplementary Fig. 13). On the other hand, the interface contains rich In-O–In structures as shown in Supplementary Fig. 14, which contributes to the formation of oxygen defects around metallic In[32]. These findings suggest that subsurface oxygen defects gather surrounding metallic In. The proportions of surface oxygen defects of $In_2O_3$ and In-Em $In_2O_3$ are estimated to be 0.032% and 0.0017%, respectively.

Then we attach importance to the local structure of the oxygen defects that exerts a substantial impact on the performance of these catalysts. Usually, it is unfavorable in thermodynamics to remove multi-oxygen atoms from the irregular octahedral $InO_6$ unit cell ($C_{2v}$ group) via thermal treatment because of the rather high energy required[26,33]. This gives rise to the feature of O-penta-coordinated state in $In_2O_3$ consisting of one central In, five coordinated O, and one oxygen vacancy with two electrons which are paired most frequently in one general unit, being the most stable conformation of $In_2O_3$[33]. The O-penta-coordination mode shackles an electron-transfer event since local electrons are imprisoned by a strong electrical field around oxygen defect[8]. With more coordinated O atoms removed via reduction, In atom

clusters emerge in one general unit around oxygen defects, with more In–O–In attached on metallic In for enhanced electronic interaction between $In_2O_3$ and metallic In (Supplementary Fig. 14). Moreover, more than two electrons are required to compensate local non-balance charge around oxygen defect induced by $In^0$ atom cluster (Supplementary Fig. 15). This $In^0$-mediated oxygen defect structure would increase the number of free electrons in In-Em $In_2O_3$ compared with $In_2O_3$, as verified by electron spin resonance (ESR) spectra (Fig. 2c). Obviously, the ESR signal of In-Em $In_2O_3$ demonstrates a 1.6-fold enhancement related to $In_2O_3$ under light irradiation. The enhancement of In-Em $In_2O_3$ (2.1-fold) is greater in the dark than under light irradiation, suggesting that In-Em $In_2O_3$ possesses more unpaired electrons ($z > 2$) unbound dominantly in defect states. The ESR increase is attributed to $In^0$-induction spin enhancement originating from electric field polarization around oxygen defect.

Positron annihilation spectroscopy (PAS) is a useful tool to unravel the microstructure of the catalysts, where the lifetime and intensity of the positrons relate to the size, relative content, and distribution density of oxygen defects[34]. The results from this analytical method were fitted best with three-lifetime components and the lifetimes and relative intensities of the positrons for $In_2O_3$ and In-Em $In_2O_3$ were presented in Fig. 2d and Supplementary

Table 3. The small amount of the third component arises from ortho-positron annihilation inside a few large voids (defect clusters or micropores) in the catalysts[35]. Compared with $In_2O_3$, In-Em $In_2O_3$ displays a longer third lifetime ($\tau_3$), reflecting defect accumulation at the interface. The first component is attributed to free annihilation of the positrons by a bulk state in a crystal[36,37]. The dissimilarity of the first lifetimes ($\tau_1$) between $In_2O_3$ and In-Em $In_2O_3$ is only 9 picoseconds, attributable to small vacancies[38,39] or shallow positron traps[40] in the bulk. Such a tiny distinction also indicates their similar isolated vacancy structures (the form of O–In-(O)$V_o$-In–O which could be distributed on the surface and in the bulk of $In_2O_3$ but predominantly exist in the bulk of In-Em $In_2O_3$, on account of the previous catalyst characterization on the marked difference of sub- and surface oxygen defects of $In_2O_3$ and In-Em $In_2O_3$. The second component originates from the trapping of free positrons by larger-size defects[41]. In-Em $In_2O_3$ presents a much higher second lifetime ($\tau_2$) than $In_2O_3$, implying that $In_2O_3$ includes oxygen-vacancy associates on the surface[35] whereas larger-size defect complexes exist in In-Em $In_2O_3$, such as a metal-mediated defect complex in the form of O–In-(O)$V_o$-In–In (Fig. 2e).

**TOF activity over In-Em $In_2O_3$ is 866 times higher than that over $In_2O_3$ under light irradiation**. The photothermocatalytic $CO_2$ reduction was conducted under light irradiation. Under the molar ratio of $H_2/CO_2/Ar$ = 9/3/8 and pressure at 0.18 MPa, CO was the main product with a selectivity of 99.99%. The reaction time of $CO_2$ reduction for calculating TOF and the mass and area-specific activity as follows is 1 h unless a special reaction time is mentioned. For apparent activity, two types of catalyst evaluation indexes were considered: mass-specific activity normalized by the mass of catalyst and area-specific activity by the surface area of the catalyst. The mass and area-specific activity over In-Em $In_2O_3$ (380 °C) are 2.7-fold (8.6 vs. 3.2 mmol $g^{-1}$ $h^{-1}$) and 40.0-fold (1.2 vs. 0.03 mmol $m^{-2}$ $h^{-1}$) higher than that over $In_2O_3$ (310 °C), respectively (Supplementary Fig. 16), at the same light intensity (8.2 W $cm^{-2}$). As shown in Supplementary Fig. 17, In-Em $In_2O_3$ exhibited a stronger light-to-heat conversion in relation with $In_2O_3$. To eliminate the influence of different temperatures caused by the same light irradiation, the activities were measured at the identical reaction temperature. Their mass-specific activities are almost equivalent while the area-specific activities exhibit a difference of one order of magnitude (17-fold) (Supplementary Fig. 16). However, the apparent activity fails to reflect the exact essence of active sites; hence we evaluated the performance in terms of one active site. The $CO_2$ adsorption site (oxygen defect) of $In_2O_3$ definitely acts as active site of $CO_2$ reduction[7,21–23,26]. From $CO_2$-TPD, we derived the adsorbed $CO_2$ amount to estimate the molar quantity of active sites. Accordingly, there exists an incredible distinction between $In_2O_3$ and In-Em $In_2O_3$ in the TOF activities (mass-specific activity/number of active sites, see methods) under identical light intensity (866-fold) and identical reaction temperature (376-fold) (Fig. 3a). The reason for this obvious difference will be discussed in the following section. Moreover, TOF activity of CO production over the catalyst surpasses that over most of the reported catalysts (Supplementary Table 4).

The TOF activity over In-Em $In_2O_3$ turns out to be higher than that over $In_2O_3$. But the active sites for the calculations only correspond to ones before reaction. As we know, oxygen defects could undergo transformations during catalysis: oxygen defects could be replenished by foreign oxygen from COOH intermediate dissociation; the neighboring chemical composition or the structure of oxygen defects could be altered[42]. To avoid the change in the concentration of oxygen defects and retain the structure of initial oxygen defects in the process of catalysis as far

as possible, low-temperature and short-time reactions were conducted, respectively (Supplementary Fig. 18). First, the low-temperature reaction was operated at 250 °C over 30 min and the initial rate over one active site of In-Em $In_2O_3$ becomes 557-fold faster than that of $In_2O_3$. Then, turnover numbers (TONs) were measured at 350 °C over 10 min and In-Em $In_2O_3$ exhibits 327-fold higher vs. $In_2O_3$. These results further verify that the active sites of In-Em $In_2O_3$ have much stronger catalytic function than that of $In_2O_3$ (two orders of magnitude enhancement).

We have measured the time-dependent catalytic activity over In-Em $In_2O_3$ to verify the stability of the catalyst. Under the same condition (300 °C, $H_2/CO_2/Ar$ = 9:3:8), the catalytic reaction over In-Em $In_2O_3$ was cycled for ten runs (the reaction time of each run was 1 h). The result shows that the catalytic cycling performance over In-Em $In_2O_3$ is stable (Fig. 3b). The content, structure and composition of oxygen defects of the spent catalyst were revealed by the characterization measurement for the nature of performance stability (Supplementary Fig. 19 and Supplementary Table 5).

The improvement of TOF activity is closely related to the microstructure of the catalyst, such as the transition of composition and phase. The formation of metallic In in $In_2O_3$ is the most significant change between $In_2O_3$ and In-Em $In_2O_3$ in their microstructures. Therefore, the correlation between the catalyst performance and metallic In was investigated. We measured a suite of activities over $In_2O_3$ at different reaction temperatures and obtained the initial curve ($CO_2$ conversion vs. T, dashed line) and the derived curve (ln k vs. 1000/(T + 273.15), k is rate constant, solid line) in Fig. 3c. Based on the Arrhenius equation which describes the relationship between the reaction rate constant and reaction temperature, the change in the apparent activation energy can be reflected by the slope of the derived curve. If the derived plot is a straight line, its apparent activation energy remains. But in fact, it is a curve, signifying that the active component around the oxygen defects of $In_2O_3$ varies with reaction temperatures and a new phase (metallic In) in $In_2O_3$ appeared upon elevating reaction temperatures. The high-temperature part of the curve is consistent with the apparent activation energy over In-Em $In_2O_3$. Moreover, this suggests that the catalyst underwent distinctly different reaction pathways[43]. Indeed, after a high-temperature reaction, a part of the $In_2O_3$ was reduced to metallic In (Supplementary Fig. 20). The result indicates that metallic In modifies the structure of the oxygen defects and engages in the TOF activity enhancement.

The transition-state theory can describe the reactivity of the catalyst, and the changes in enthalpy ($\Delta H_{\neq}$) and entropy ($\Delta S_{\neq}$) of the transition state are given by the Eyring formula (the plot of ln (k/T) vs. 1/T) for the rate-limiting step. $\Delta H_{\neq}$ and $\Delta S_{\neq}$ on $In_2O_3$ at the low and high temperature were calculated to $\Delta H_{\neq}$ (18.1 and 78.6 J $mol^{-1}$) and $\Delta S_{\neq}$ (−179.3 and −69.4 J $mol^{-1}$ $K^{-1}$), respectively. The transition-state complex is penalized by the negative entropy, but is strongly chemisorbed by oxygen defects. In addition, the reaction underwent different activated complexes. The complex is dominantly controlled by entropy because of $H/TS$« 1, and the entropy reflects the bound state of the transition state at the active site. The much more favorable entropy change in high-temperature reaction indicates that the transition state is in a more disordered state, driving its transformation to products more readily. Meanwhile, this also gives rise to a lower Gibbs free energy of activation at a constant temperature.

In addition, we adopted mild hydrogen peroxide to oxidize the surface of metallic In of In-Em $In_2O_3$ (named as In-Em $In_2O_3(H_2O_2)$), forming a thin layer of $In_2O_3$ with more surface oxygen defects. This can create more surface oxygen defects around metallic In. The performance evaluation displays the enhanced activity over In-Em $In_2O_3(H_2O_2)$ compared with

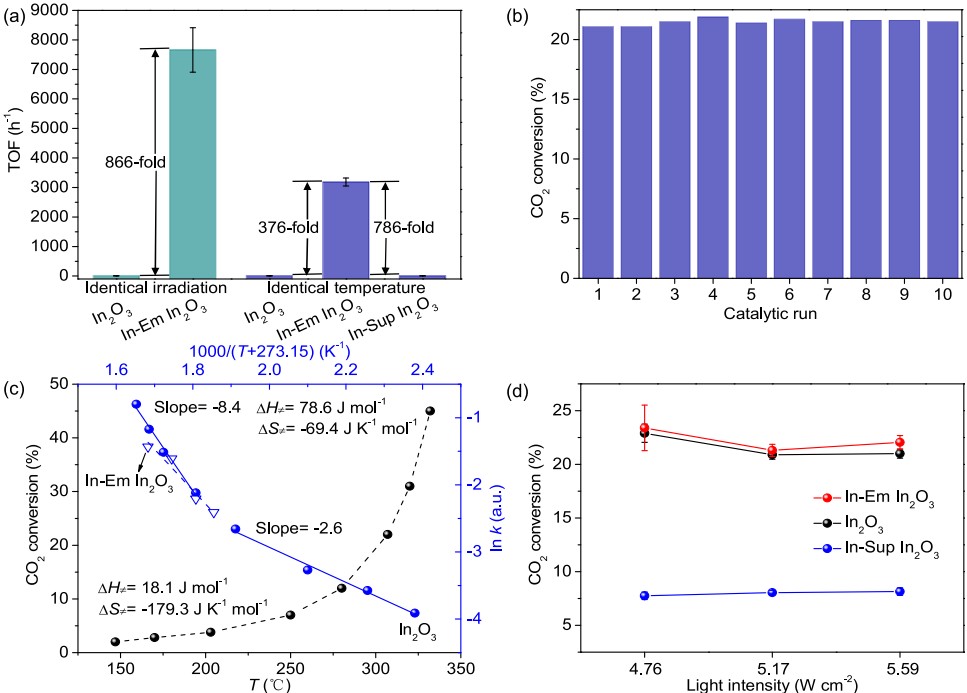

**Fig. 3 CO₂ reduction performances over 1 h. a** TOF activities over In₂O₃, In-Em In₂O₃ and In-Sup In₂O₃. **b** Cycling experiment over In-Em In₂O₃ for 10 runs. **c** Reaction temperature dependence of CO₂ conversion, the corresponding curve of ln $k$ ($k$: rate constant) vs. 1000/($T$ + 273.15) and corresponding $\Delta H_{\neq}$ and $\Delta S_{\neq}$ extracted from Eyring plot. **d** CO₂ conversion over In₂O₃, In-Em In₂O₃ and In-Sup In₂O₃ under light irradiation. Note that insufficient temperature was compensated via electric heating to 300 °C. (The error bars represent standard deviation).

In-Em In₂O₃ (Supplementary Fig. 21). The characterization (Supplementary Fig. 22) suggests that the overall structure of In-Em In₂O₃(H₂O₂) changes little, but the content of surface oxygen defects around increases. This further verifies that the oxygen defects around metallic In exhibit the stronger reactivity.

To eliminate the role of metallic In as a co-catalyst, metallic In supported In₂O₃ (In-Sup In₂O₃) was prepared[44]. XRD patterns show that the catalyst contains In₂O₃ and metallic In and the amount is similar with that of In-Em In₂O₃ (Supplementary Fig. 23). The 2θ of In-Sup In₂O₃ moves to the lower, suggesting a strong interaction between metallic In and In₂O₃ in In-Sup In₂O₃. In-Sup In₂O₃ does not contain other foreign substances with washed by large amount of water/alcohol (Supplementary Fig. 24). The activities over In₂O₃, In-Sup In₂O₃ and In-Em In₂O₃ were compared at the same temperature (Fig. 3d) and the result suggests that there is no obvious change in their activities upon changing the light intensity, consistent with Supplementary Fig. 25, corroborating the slight effect of the photocatalysis and charge transfer on CO₂ reduction. If metallic In acts as the photogenerated electron separator, with light irradiation, more electrons are injected into metallic In and the catalyst performance would be improved. However, the activity over In-Sup In₂O₃ is lower than that over In₂O₃ (Fig. 3d), thus the effect of electron promoter of metallic In is excluded. Moreover, compared with In-Em In₂O₃, the activity over In-Sup In₂O₃ is significantly decreased under light irradiation (Fig. 3d), despite the markedly higher CO₂ adsorption for In-Sup In₂O₃ (Supplementary Fig. 26), indicating that metallic In is not the co-catalyst for favoring the dissociation of C–O of CO₂ or COOH intermediate. As expected, TOF activity over In-Sup In₂O₃ is so much lower than that over In-Em In₂O₃ (Fig. 3a), suggesting that the simple In-supporting structure cannot improve intrinsic activity.

Photothermal conversion over the system is one of the key factors dictating the photothermocatalytic performance of

CO₂ reduction. The photothermal conversion capability of In₂O₃ is rather low (Supplementary Fig. 17a) because of the light absorption of the wavelength below 500 nm and dominant radiative emission. Compared with In₂O₃, In-Em In₂O₃ displays a more efficient photothermal conversion due to full-spectral light absorption (to near-infrared light) and a high probability of nonradiative relaxation (Supplementary Fig. 17). The photothermal effect of In-Em In₂O₃ originates from oxygen defects and light absorption characteristics of metallic In. As reported, oxygen defects can create mid-gap energy state and thus increase light-to-heat conversion due to enhanced light absorption[7] and "trap-assisted recombination"[45]. Compared with oxygen defects, metallic In of In-Em In₂O₃ plays a dominant role in light-to-heat conversion which heats up the metal lattice by electron–phonon scattering. Due to superior thermal conduction of metallic In, the concentrated energy in metallic In is then rapidly transferred to the active site of In₂O₃ portion for CO₂ reduction via phonon-phonon relaxation[46]. However, in the future, it is worth investigating which of the light absorption modes from metallic In exhibits the highest efficiency of light-to-heat conversion, including interband-transition absorption, intraband-transition absorption, and plasmon-resonance absorption[45–47].

**Electron delocalization among oxygen defects**. The capability to activate CO₂ over the catalyst is closely linked with the electron density of the active sites which is correlated with electron transfer. Variable temperature ESR spectroscopy (Fig. 4a and Supplementary Fig. 27) can be used to analyze the kinetic behavior of electron transfer in oxygen defect at the interface between In₂O₃ and metallic In. The dynamics includes two processes: (a) electron injection from In₂O₃ to metallic In and (b) electron migration from the subsurface to the surface. At a constant temperature of 100 K, the ESR signal was tested in the dark and

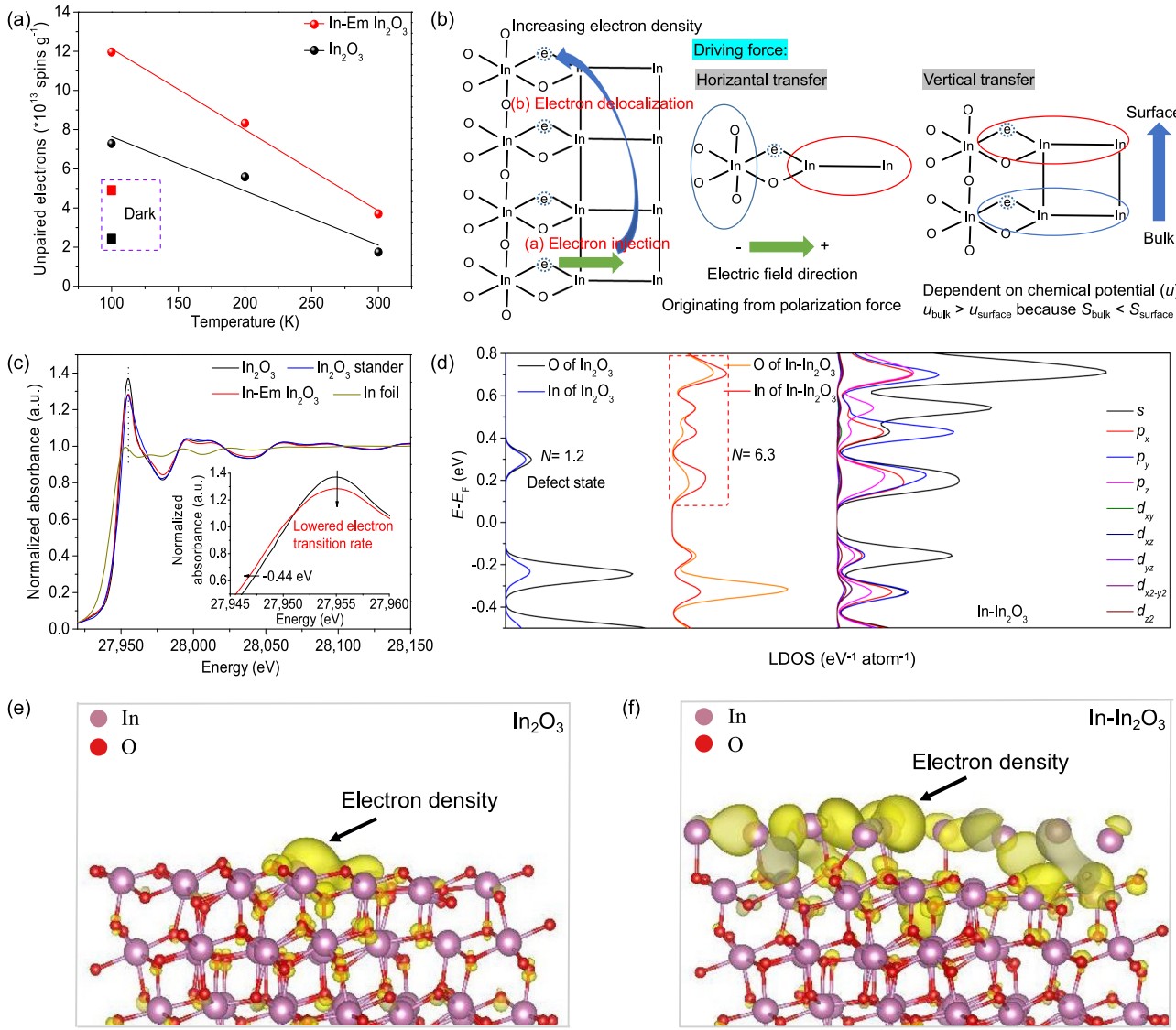

**Fig. 4 Electron delocalization among oxygen defects. a** Temperature-dependent change of unpaired electrons in $In_2O_3$ and In-Em $In_2O_3$ at 100, 200, and 300 K under light irradiation and numbers of unpaired electrons at 100 K in the dark. **b** The schematic presence of electron transfer at the interface and its driving forces. **c** Normalized In K-edge XANES spectra of $In_2O_3$ and In-Em $In_2O_3$. **d** LDOS of In–$In_2O_3$ near the Fermi level (*N*: number of energy levels) and orbital contribution in LDOS of In–$In_2O_3$. Charge distribution derived from the wave function of the energy levels at a CBM of (**e**) $In_2O_3$ and (**f**) In–$In_2O_3$.

under light irradiation, respectively. In the dark, the ESR signal of In-Em $In_2O_3$ is twice that of $In_2O_3$. At this time, the unpaired electrons in the catalyst are in an equilibrium state, and the population of unpaired electrons of In-Em $In_2O_3$ is higher than that of $In_2O_3$. The previous characterization demonstrates that the sum of their oxygen defects appears the same, and in general, each oxygen defect contains two electrons[4]. But as for In-Em $In_2O_3$, the average number of electrons per oxygen defect is apparently greater than two, implying the sharing of electrons among oxygen defects. Irradiation to the catalyst produces some non-equilibrium electrons, of which the population in In-Em $In_2O_3$ is 2.1 times higher, related to $In_2O_3$. Under the interface polarization force, some of the non-equilibrium electrons generated in the $In_2O_3$ phase are injected into the metallic In, and the O–In–$V_O$(O)-In–In structure acts as an electron "bridge" (Fig. 4b).

The ESR signal shows temperature-dependent decay, mainly due to the increase in electron conduction resistance with temperature increasing. In-Em $In_2O_3$ exhibits a faster ESR signal decaying rate compared with $In_2O_3$ (Fig. 4a), implying that

unpaired electrons are transported along a different approach in the In-Em $In_2O_3$; actually, the unpaired electrons of In-Em $In_2O_3$ move along metallic In after electron injection, because the resistance of metallic In increases faster with a temperature rising compared with $In_2O_3$, which results in a faster ESR signal decay in In-Em $In_2O_3$. Surface oxygen defects are affected by adsorption state and surface state. The number of surface microscopic states turns out to be greater than that of the bulk counterpart, that is, entropy (*S*) divergence of $S_{surface} > S_{bulk}$, therefore, the chemical potential of the bulk phase is often higher than that of the surface phase, driving the electrons in subsurface oxygen defects to transport to surface counterparts (Fig. 4b), thereby incrementing the number of unpaired electrons in active sites.

The migration of electrons from the subsurface to the surface is affected by a relatively large resistance in the oxide semiconductor, causing them to be annihilated during the movement through electron-hole recombination, electron–phonon and electron–electron interaction. The mean free path of electrons in oxide semiconductors is less than 5 nm regardless of bound force from metallic ions around, thus only surface electrons would participate in the reaction. The

metallic In greatly reduces the resistance to electron migration through the lattice from the subsurface to the surface (Maximum conductivity multiple is 12 orders of magnitude compared with semiconductors[48].) and increases the mean free path of electrons. Without feeding with reactants, the electrons stay in equilibrium between the surface and the subsurface. Due to the existence of the surface state, the electron concentration on the surface presents generally higher than that of the subsurface. The surface/subsurface electron-density ratio of In-Em $In_2O_3$ is greater than that of $In_2O_3$ due to the enrichment of electrons via metallic In. When the active site interacts with the reactant electronically, the electron density on the surface decreases, and the redistribution of the subsurface electrons is required to maintain a dynamic equilibrium state.

Electronic characteristics of oxygen defects were also explored via PAS. The positron trapping rates were calculated, which is proportional to the number of defects and density of negative charges in an individual defect. The positron trapping rate is directed against the second component exclusively pertaining to defects[37]. As shown in Supplementary Figs. 28 and 29, though $In_2O_3$ surpasses In-Em $In_2O_3$ in the positron trapping rate, the trapping rate constant over In-Em $In_2O_3$ is higher, implying that the defects of In-Em $In_2O_3$ have a stronger capability to trap positrons, namely, more negative charges. Besides, the electrons in the defects of In-Em $In_2O_3$ display higher delocalization energy (Supplementary Fig. 30). Therefore, different from electrons trapped in the oxygen defects of $In_2O_3$, electrons in the oxygen defects of In-Em $In_2O_3$ are shared by each other at the interface.

Metallic In is responsible to electron delocalization at the interface and more essentially, the O–In-(O)$V_o$-In–In structure of In-Em $In_2O_3$ makes contribution. Therefore, the X-ray absorption near edge structure (XANES) and extended X-ray absorption fine structure (EXAFS) spectra were measured. In the XANES spectra (Fig. 4c), the tiny energy difference (0.44 eV) of the K-edge absorption at 27922 eV between $In_2O_3$ and In-Em $In_2O_3$ suggests that the crystal network of In-Em $In_2O_3$ remains. Moreover, compared with the $In_2O_3$ stander, their K-edge absorption is shifted to the lower energy, attributed to a lower average oxidation state of In species which corresponds to a lower coordination around In atoms, namely, oxygen defects[7,49,50]. Metallic In of In-Em $In_2O_3$ does not exhibit obvious electronic feature of In foil, because of the similar In–In bonding lengths of $In_2O_3$ and metallic In and low sensitivity to low ordered structure of metallic In[51]. The intensity of the "white line" of In species in In-Em $In_2O_3$ is lower than that in $In_2O_3$ (Fig. 4c, inset), attributable to a slower electron transition[52] from the $1s$ orbital to unoccupied N4,5 states. The electron transition needs to reach certain energy. The reduction of the electron-transition probability is because the delocalization of the electron reduces the population of high-energy electrons. The R and K space EXAFS curves and wavelet transform analysis are shown in Supplementary Figs. 31–34. The fitting of the EXAFS result (Supplementary Table 6) gives a lower coordination number and higher Debye-Waller factor for In-Em $In_2O_3$, implying a more disordered surface structure. The In–In coordination for In-Em $In_2O_3$ could correspond to the $dsp^2$ hybrid at the interface whereas the lattice region in $In_2O_3$ predominantly to the $d^2sp^3$ hybrid for the octahedral unit. Obviously, In-Em $In_2O_3$ possesses more $s$-orbital and less $d$-orbital components, and consequently, the overlapping degree of In–In orbitals turns greater due to high dispersion of $s$-orbital. Therefore, In-Em $In_2O_3$ exhibits a stronger electron-delocalization effect at the interface. Moreover, the O–In-(O)$V_o$-In–In structure partly contributes to the In–In shell and thus facilitates a larger extent of electron delocalization.

Density functional theoretical (DFT) calculations were carried out to verify the electron delocalization. Local density of states (LDOS) of defective $In_2O_3$ (Supplementary Fig. 35a) presents a small LDOS defect state dominated by an unoccupied $In_{5s}$ level in the bandgap. LDOS in Supplementary Fig. 35b displays metallic continuity behaviors of metallic In that can match well with defect state and delocalize electrons of defect state[53,54]. Upon interaction of $In_2O_3$ with metallic In of low coverages, the Fermi level of $In_2O_3$ shifts to higher energy and a new free-electron-like band appears near Fermi level[30]. On the $In_2O_3$ slab with one oxygen vacancy, some In atom clusters were constructed (donated as In–$In_2O_3$) for LDOS calculation. The overall energy band of In–$In_2O_3$ remains unchanged (Supplementary Fig. 36), complying with the XANES and EXAFS results. However, only one defect state (0.30 eV) appears in the bandgap of $In_2O_3$ while In–$In_2O_3$ has continuous defect states (Fig. 4d) which suggests the shareability of electrons of oxygen defects. Integrating LDOS(E) from the band bottom to the Fermi level can get the electron occupied energy levels ($N$) in the defect state, $N_{In2O3}$ = 1.2 and $N_{In-In2O3}$ = 6.3, so the corresponding filling electron number is $N_{In2O3}$ = 2.4 and $N_{In-In2O3}$ = 12.6, respectively. This defect state belongs to the $s$–$p$ band and exhibits a good electron delocalization between defect states (Fig. 4d). In addition, the charges in the valence band maximum (VBM) and the conduction band minimum (CBM) of $In_2O_3$ are restricted in the oxygen defects, while the charges in the CBM of In–$In_2O_3$ demonstrate more extensive distribution (Fig. 4e, f and Supplementary Fig. 37). Consequently, the active sites on the surface converge more available electrons from subsurface oxygen defects to ensure successful $CO_2$ activation.

**Enhanced $CO_2$ adsorption and activation.** To verify the existence of the electron exchange between the active sites of In-Em $In_2O_3$ and $CO_2$, $Fe^{3+}$ salt was selected as an electron scavenger. As expected, the grinding mixture of In-Em $In_2O_3$ and $Fe^{3+}$ salt markedly attenuated its photothermocatalytic and thermocatalytic performances (Supplementary Fig. 38), confirming that the electrons are key active species for $CO_2$ reduction. At room temperature, $CO_2$ adsorption over $In_2O_3$ and In-Em $In_2O_3$ was measured through Fourier transform infrared (FT-IR) spectroscopy (Supplementary Fig. 39). Enhanced adsorption of $CO_2$ on In-Em $In_2O_3$ is observed (Fig. 5a) and $CO_2$ adsorption band shifts from 1304.5 $cm^{-1}$ for $In_2O_3$ to 1286.3 $cm^{-1}$ for In-Em $In_2O_3$ because of more electrons delivered to the anti-bond orbital of $CO_2$. The peaks at 3500–3800 $cm^{-1}$ in FT-IR spectra were chosen as the standard of the $CO_2$ adsorption amount, which are assigned to a combination mode ($v_1 + v_3$) of the adsorbed $CO_2$[55]. In Fig. 5b, In-Em $In_2O_3$ performs a significantly faster $CO_2$ adsorption rate compared with $In_2O_3$ and reached chemical equilibrium within 110–130 s. DFT calculations were carried out to further unravel the electronic interaction between the active sites and the reactants/intermediates. The binding configurations of each adsorbate on the surfaces of $In_2O_3$ and In–$In_2O_3$ were obtained through theoretical structural optimization (Supplementary Fig. 40, 41), respectively. Adsorption energies were employed to evaluate the strength of the electronic interaction. By contrast with $In_2O_3$, all the adsorption energies over In–$In_2O_3$ are more negative especially H uptake (Fig. 5c), indicating facile $CO_2$ hydrogenation and COOH dissociation at the active sites via consecutive electron-transfer events. The stronger interaction prolongs the residence times of the adsorbates at the interface, $10^8$-fold for $CO_2$ and $10^{29}$-fold for hydrogen higher than that on $In_2O_3$ calculated from the equation

$$\tau = \tau_0 \exp(-E_{ads}/RT) \qquad (1)$$

where $\tau$ is the lifetime of adsorbate, $E_{ads}$ is adsorption energy, R is the molar gas constant, $T$ is temperature and $\tau_0$ is the lifetime of a surface vibration ($\sim 10^{-13}$ s)[56]. Therefore, the steady-state

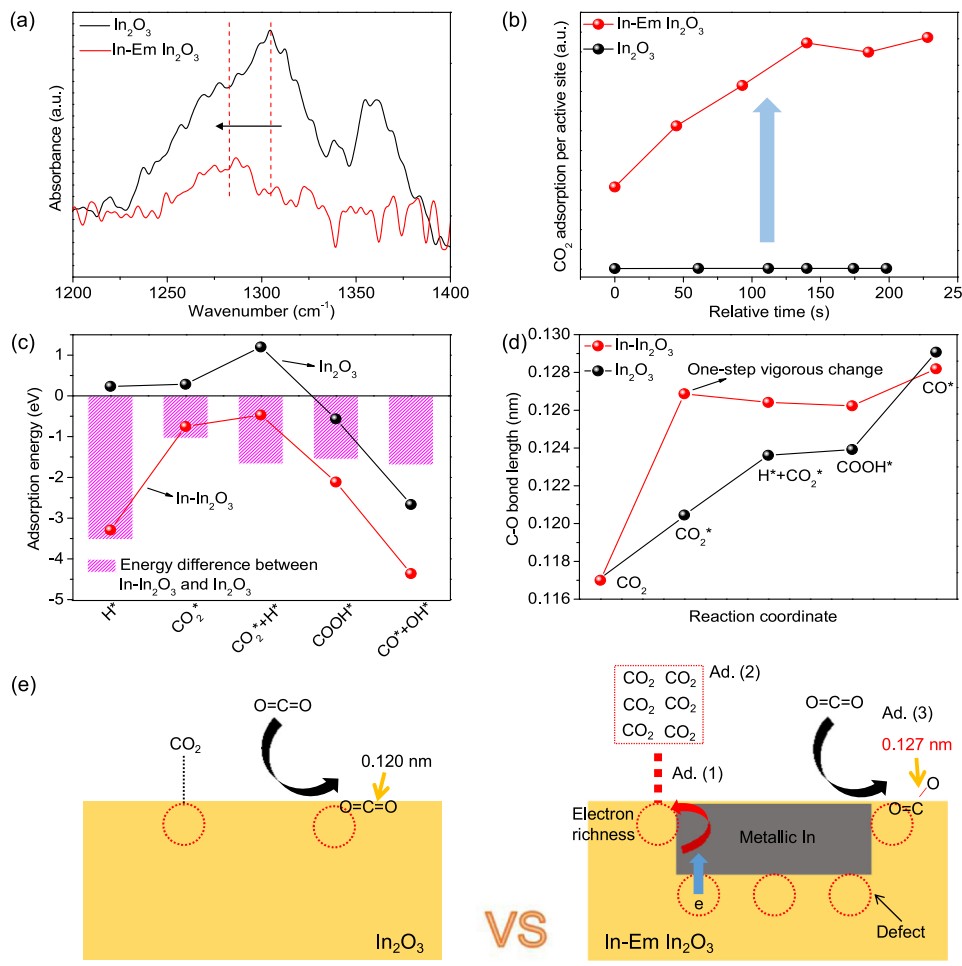

Ad. (1) Improved interaction with $CO_2$; Ad. (2) Increasing $CO_2$ steady concentration;
Ad. (3) Vigorous activation, with respect to one oxygen defect.

**Fig. 5 Enhanced $CO_2$ adsorption and activation. a** FT-IR spectra of $CO_2$ adsorption in the range of 1200–1400 $cm^{-1}$. **b** Normalized $CO_2$ adsorption amount by the number of active sites. **c** Adsorption energies on $In_2O_3$ and $In–In_2O_3$. **d** Change in C–O bond lengths over $In_2O_3$ and $In–In_2O_3$. **e** The schematic picture emphasizes three types of catalytic roles for $CO_2$ reduction over In-Em $In_2O_3$.

concentrations of $CO_2$ and H are extremely high at the interface, favoring intermolecular collision. The adsorption energies of $CO_2$ and H on metallic In were calculated to be −0.081 and −1.058 eV, respectively. The adsorption energies of $CO_2$ and H on $In–In_2O_3$ (namely at the interface between metallic In and $In_2O_3$) are much lower than that on $In_2O_3$ and metallic In, which most likely evidence that the active sites are at the interface between metallic In and $In_2O_3$. This is consistent with the conclusion above that the oxygen defects at the interface function as the active sites for $CO_2$ reduction. In addition, electron exchange capability can be deduced by the change of the C–O bond length. Over $In–In_2O_3$, the bond of $CO_2$ is elongated by 0.099 Å while it is only extended by 0.034 Å over $In_2O_3$ (Fig. 5d), indicating the stronger electronic interaction with $CO_2$ over $In–In_2O_3$. Noteworthy, only one step is required to vigorously activate the adsorbed $CO_2$ into COOH over $In–In_2O_3$, whereas it takes multiple steps of energy conversion to convert $CO_2$ to CO over $In_2O_3$. The advantages (Ad.) of electron richness from electron delocalization of subsurface oxygen defects are illustrated in Fig. 5e.

The reaction pathway in the system is clarified including the roles of heat, photogenerated carrier and $H_2$ in $CO_2$ reduction. For the present photothermocatalysis, the main contribution comes from the thermochemical pathway generated by light irradiation while the photochemical pathway makes minor contribution (Supplementary Fig. 25). Therefore, the catalytic reaction is called light-induced thermocatalysis[47]. Here, a population of photons are absorbed by metallic In portion and oxygen defects (via "trap-assisted recombination") and converted into thermal energy, respectively. The thermal chemistry facilitates the transfer of charge carrier, the excited vibration of the related species and the formation of the phonon of the ground state of $In_2O_3$, which lower the reaction barrier of $CO_2$ reduction. Simultaneously, some photons are absorbed by $In_2O_3$ portion, forming photogenerated carriers. There are two types of evolution directions for the photogenerated carriers. One way is electron-hole recombination generating more heat, which is dominant, and the other is to be transferred to $CO_2$ adsorbed. It is worth noting that most of the photogenerated charges in $In_2O_3$ portion are not delivered to $CO_2$ adsorbed specifically at the oxygen defects on the surface because of the very small chemical reaction region in spite of increasing ESR signals upon light irradiation as demonstrated in Supplementary Fig. 27. If the photogenerated electrons can interact with $CO_2$ adsorbed, the energetic electrons would be injected, generating anion species[57]. Otherwise, the photochemical pathway makes a minor contribution to the catalysis. The $H_2$ is dissociated into two H species in either a heterolytic or a homolytic way with the participation of lattice In

and $O$[58–61], which binds with $CO_2$ to form COOH or binds with OH from the dissociation of COOH to form $H_2O$, respectively[7]. However, the atmosphere including $H_2$ would not increment the number of surface oxygen defects of $In_2O_3$ and the amount of metallic In during the catalytic process, as indicated by $H_2$-TPR pattern (Supplementary Fig. 2) and XRD pattern (Supplementary Fig. 19a).

In summary, a strategy of extracting electrons in subsurface oxygen defects for $CO_2$ reduction was reported via constructing In-Em $In_2O_3$ nanoflake with metallic In embedded. The oxygen defects of In–$In_2O_3$ are distributed at the interface between $In_2O_3$ and metallic In and comprise large-size defect complexes featuring the basic structure of $O–In-(O)V_o-In–In$. In-Em $In_2O_3$ exhibits remarkably higher TOF activities than $In_2O_3$ under light irradiation. The $O–In-(O)V_o-In–In$ structure at the interface engenders delocalization of electrons in the subsurface oxygen defects mediated by metallic In that greatly increases the electron density of the active sites, facilitating electron exchange between the active site and $CO_2$ and stabilizing COOH intermediate. The study helps to understand the active sites of $In_2O_3$ and paves the way to develop new catalysts and improve catalyst performance involving oxygen defects.

## Methods

**Preparation of $In_2O_3$.** $In(NO_3)_3$•$4H_2O$ (0.013 mol, 5.0 g) and urea (0.039 mol, 2.4 g) were dissolved in deionized water (600 mL), followed by magnetic stirring for 15 min. The obtained solution was transferred into a Teflon-lined autoclave. The autoclave was sealed and heated at 140 °C for 16 h. After cooling down, the suspension was centrifuged and washed with deionized water. The white solid was dried overnight under vacuum at 60 °C and then underwent calcination in a muffle furnace at 300 °C (with the temperature-ramp rate of 7~8 °C min$^{-1}$) for 3 h to produce a yellow powder.

**Preparation of In-Em $In_2O_3$.** After grinded with a mortar, the virgin $In_2O_3$ was placed in a porcelain boat without a lid and underwent calcination at different times or temperatures in $H_2$/Ar (1/9) atmosphere with a temperature-ramp time of 20 min.

**Preparation of In-Sup $In_2O_3$.** According to the literature[44], highly dispersed metallic In in $H_2O$ was prepared first. $InCl_3$•$4H_2O$ (2.5 mmol, 735 mg) and disodium citrate hydrate (1.9 mmol, 500 mg) were dissolved in diethylene glycol (100 mL) in a three-necked flask. Under $N_2$ protection and vigorous stirring, the solution was heated to 100 °C. Subsequently, $NaBH_4$ (25.0 mmol, 945 mg) was dissolved in deionized water (2 mL) and added to the solution which finally turns dark brown. The reaction continued to be stirred for 5 min at 100 °C. After cooling down, the suspension was centrifuged and washed with a large amount of alcohol and deionized water. The obtained precipitate was dispersed in deionized water (30 mL) and $In_2O_3$ (300 mg) was added. After stirring for 30 min, the suspension was centrifuged and washed with a large amount of deionized water. The gray solid was dried overnight under vacuum at 60 °C.

**Preparation of In-Em $In_2O_3(H_2O_2)$.** 200 mg of In-Em $In_2O_3$ was added into 1% diluted $H_2O_2$ solution. After ultrasonic treatment, the dispersion was stirred for 3 h, followed by centrifugation and drying under vacuum at 60 °C.

**Catalyst characterization.** The morphologies of the catalysts were characterized by FE-SEM (JEOL JEM-6700F) at a working voltage of 8 kV, TEM (HITACHI H-7000FA, 100 kV), high-resolution TEM (JEM 2100 F) and AFM (BRUKER Dimension Icon). XRD patterns were recorded on an X-Pert diffractometer (BRUKER D8 ADVANCE) equipped with graphite monochromatized Cu-K$_\alpha$ radiation. XPS were obtained on a ThermoFisher EscaLab 250Xi using monochromatic Al K$\alpha$ source (Ephoton = 1486.6 eV) with 10 mA filament current and 14.7 keV filament voltage source energy spectrometer (Correction value of $C_{1s}$ in the XPS spectra was 284.7 eV.). $CO_2$-TPD and $H_2$-TPR curves were carried out on an Auto Chem II2920 chemisorption apparatus with a temperature-ramp rate of 10 °C min$^{-1}$ after pretreatment at 200 °C for 0.5 h in Ar. Specific surface area and pore size distribution were measured through a high-speed automated surface area and pore size analyzer (TriStar II 3020 V1.03.01) using the multipoint Brunauer-Emmet-Teller (BET) analysis method. Absorption spectra were analyzed through an Agilent Cary60 spectrophotometer and steady/transient fluorescence spectra were measured with a FLS1000 fluorescence spectrometer. Raman spectra were acquired using a Thermo Scientific DXR Raman Microscope at the laser excitation wavelength of 780 nm and an intensity of 20 mW. TG analysis was performed on a Mettler Toledo TGA/DSC 1STAR$^e$ system (gas flow rate: 15 mL min$^{-1}$,

temperature range: from 50 to 900 °C, temperature-ramp rate: 10 K min$^{-1}$). ESR data were collected using a Bruker EMXmicro-6 X-band spectrometer. PAS was measured via the apparatus DPLS3000 (the size of the sample film is $12 \times 12 \times 2$ mm$^3$). XANES and EXAFS spectra at In K-edges were recorded at the XAS station (BL14W1) of the Shanghai Synchrotron Radiation Facility using the method given in the literature[62].

**Performance evaluation.** Photothermocatalytic $CO_2$ reduction in the presence of $H_2$ was conducted in a sealed batch-type reaction system. After air evacuation of the reaction vessel, the mixed feed gases containing $H_2$, $CO_2$, and Ar with the molar ratio of $H_2$/$CO_2$/Ar = 9/3/8 were introduced. The photothermocatalytic $CO_2$ reduction proceeded under light irradiation equipped with a 300 W Xe lamp (Beijing Perfectlight Technology Co., Ltd. PLS-SXE-300DUV) over 1 h. For all of these experiments, 100 mg of samples were weighed and spread onto a round shape air-permeable quartz fiber filter. The quartz fiber filter film was fixed on the stage of the reactor. The tip of the thermometer was maintained an intimate contact with the sample. The initial pressure in the reactor was kept at 0.18 MPa. The reaction temperature can be adjusted by changing the magnitude of the current of Xe lamp: first, adjust the light-irradiation current to 16 A to increase the reaction temperature; second, finely tune the position of the reactor to make the stable temperature reach the maximum; third, slowly increase the current to make the reaction temperature reach 300 °C. The adjustment time was controlled within 5 min. The reaction gas in the reaction system was collected and measured through a gas chromatograph (Agilent 7890B) equipped with a combination of Porapak Q, Molsieve 5 Å columns, and a thermal conductivity detector which can detect $CO_2$, $O_2$, $N_2$, $CH_4$, and CO.

$$\text{Mass specific activity} = \text{CO production/mass of catalyst} \qquad (2)$$

$$\text{Area specific activity} = \text{CO production/specific surface area of catalyst} \qquad (3)$$

$$\text{TOF activity} = \text{CO mass specific activity/number of active sites} \qquad (4)$$

Where the CO production refers to one over 1 h.

## Data availability

The data that support the findings of this study are available from the corresponding authors upon reasonable requests.

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

## Acknowledgements

We acknowledge the financial support from the National Natural Science Foundation of China (No. 21972052). We acknowledge Prof. Tierui Zhang for precious advice of the original idea. We acknowledge Yanbiao Shi for in-situ FT-IR measurement and Jianwei Wang for theoretical calculations. We thank Prof. Jian Wang for comments and manuscript polishing. S.O. thanks for the financial support from the "Guizi Scholar" Program of Central China Normal University.

## Author contributions

W.W. conducted all the experiments. W.W. and S.O. developed the idea and wrote the manuscript. S.O. directed the project and provided guidance for the experimental and theoretical work. Z.W., R.L., Z.L., R.S., Y.Q., D.P., and H.Y. conceived and performed some of the experiments and provided advice for the experiments. All the authors discussed the results and commented on the manuscript.

## Competing interests

The authors declare no competing interests.
