## [Peer Review File · Nature Communications]

Title: Subsurface Oxygen Defects Electronically Interacting with Active Sites on In₂O₃ for Enhanced Photothermocatalytic CO₂ ReductionREVIEWER COMMENTS

Reviewer #1 (Remarks to the Author):

In this manuscript, photothermocatalytic reduction of CO₂ into CO with high selectivity was achieved on In₂O₃ catalyst. More importantly, the catalytic activity for CO₂-to-CO conversion was dramatically increased by embedding metallic In on In₂O₃. The synergy between subsurface oxygen defects and metallic In on the surface of In₂O₃ was proved to be responsible for the improved catalytic CO₂ reduction performance. This study is interesting. It is recommended to be published on Nature Communications if the following issues can be properly addressed.

(1) What I most concerned is the stability of the catalyst. Based on the Methods part, the In-Em In₂O₃ sample was prepared by treating In₂O₃ in H₂/Ar (1/9) atmosphere, while the CO₂ reduction reaction was carried out in an atmosphere containing a H₂ with a much higher concentration (H₂/CO₂/Ar = 9:3:8) at 300°C. Based on the Supplementary Fig. 2, surface reduction of In₂O₃ can occur at over 200°C, thus, it is reasonable to speculate that the reduction of In-Em In₂O₃ by H₂ should proceed in catalytic CO₂ reduction process. This might produce much more O defects (or metallic In) and change the structure of the catalysts. Furthermore, it is known that metallic In has a low melting point that is below 200°C. Does the melting of the embedded In occur under the photothermocatalytic reaction conditions? If this happens, does it affect the catalytic activity of In-Em In₂O₃? Therefore, the time dependent catalytic activity of In-Em In₂O₃ should be performed in a much long time in order to prove the good stability of the catalyst (the time dependent activity was only performed for 30 min as shown in Supplementary Fig. 17). After this long time reaction, the structure, composition of the catalyst should be identified.

(2) Based on Supplementary Fig. 21, it can be seen that the CO₂ reduction activity of the In-Em In₂O₃ catalyst should be decided by temperature rising effect induced by the incident light, and the photocatalysis plays little effect on the CO₂ reduction especially in the visible and infrared regions. On the other hand, the electron delocalization of In-Em In₂O₃ was significantly influenced by the light illumination at the same reaction temperature as proved by the ESR in Supplementary Fig. 23 (It seems that this figure is not consistent with expression of the manuscript. As indicated in the text, the ESR signal was tested in the dark and under light irradiation, respectively. However, the light and dark signals were not marked in this figure or identified in figure caption). Based on my understanding, the contributions from thermal effect and photocatalysis were not clearly discussed and identified in the whole manuscript. I suggest that a clear reaction pathway should be added in the end of the manuscript, in which the roles of heat, photoinduced charges, and H₂ in CO₂ reduction should be clearly shown.

(3) Based on the O 1s XPS spectra of In₂O₃ and In-Em In₂O₃, similar O defects were contained in the two samples. However, the colour and light absorption properties of the two sample is dramatically different. This phenomenon should be explained.

(4) Based on the In XPS spectra in Supplementary Fig. 10, only the shift of the peaks was observed after the formation of metallic In in In₂O₃. Based on my understanding, there are two types of In (metallic and oxidation states) in the In-Em In₂O₃ sample. This phenomenon should be explained.

(5) The reaction time for catalytic CO₂ reduction should be clearly indicated for calculating TOF and the mass and area specific activity.

(6) For studying the overall morphology of the sample, SEM images of In₂O₃ and In-Em In₂O₃ should be

shown.

(7) In order to objectively understand the activity of the catalyst, the CO₂ reduction activity of In-Em In₂O₃ should be also compared with the recent works on the CO₂ reduction by thermal catalysis.

Reviewer #2 (Remarks to the Author):

The work by Wei et al., examined the use of Indium Oxide-based catalysts for the photo-thermal catalytic conversion of carbon dioxide to carbon monoxide. By controlling reduction, the work examined the role of 'subsurface' oxygen vacancies in driving selective conversion. Whilst the findings are interesting and the differences in performances are notable, there are distinct holes in understanding and some contradictions in the characterisation which I believe undermine the conclusions drawn by the work. As such, I cannot recommend the work for publication.

1) The main flaw in the work is the manner in which comparisons are drawn with respect to performance of the two materials. The use of TOF as a basis for catalyst comparison, is acceptable when the active sites are well defined, for example when they exist as metal deposits on supports. In this case, the active sites were defined by using CO₂ TPDs whereby the amount desorbed was used to estimate molar quantity of active sites. Whilst oxygen vacancies are evidenced in the literature as the active sites, the CO₂ TPDs are not an accurate measure of active sites under reaction conditions. The reaction occurs in a reducing environment, with heat and light. All of these can (and likely will) lead to in situ reduction and increase in the surface active sites. Using this basis is therefore highly inaccurate and unreliable. In fact, it is even stated in the manuscript that these active sites changes (Supp. Fig. 18)

2) The surface/subsurface reduction at 400oC is stated by literature support however the measured TPDs do not indicate this. Further, TPRs of the reduced and pre-treated samples should be provided for comparison and to further understand the reduction of the materials.

3) How can the presence of surface oxygen vacancies be ruled out? These could be playing a key role along with the increased light-to-heat conversion as a function of having black coloured reduced In present.

4) The evidence of the subsurface CO₂ is unclear, only AFM is used to confirm this. Perhaps a mild passivation to oxidise the surface then measure would be helpful.

5) The conversion of carbon dioxide is small <<25%, with such small conversions, improvements in performance of 2 fold are small, particularly without error bars. In this case, error bars are essential to draw real conclusions.

Reviewer #3 (Remarks to the Author):

In this manuscript, the authors reported that subsurface oxygen defects in the In-embedded In₂O₃ catalyst is much more active (by 866 or 376 folds) than the In₂O₃ catalyst, for reductive conversion of CO₂ to CO under photothermal or thermal conditions. The authors characterized the In/In₂O₃ catalyst using XRD, HRTEM, XPS, TPD, ESR, PAS, FTIR, and XANES. The subsurface oxygen defects were studied by

combining the results of XPS, and PAS, and XANES. The electron transfer from subsurface oxygen defect was observed by ESR under light irradiation. By DFT calculations, the authors found that the electron is delocalized from the subsurface oxygen defects onto the embedded In nanoparticles, resulting in the enhanced binding of CO₂, H, and intermediates and thus significantly improved CO₂ conversion. This work is surely of great interests to the catalytic conversion of CO₂, using subsurface oxygen defects in designed catalysts. Comprehensive experimental and computational analyses were performed to support the main argument. I would suggest the manuscript to be published in the journal of Nature Communications, subjected to the following minor revision.

1. In Figure 1 caption, the descriptions of (a)-(f) do not match with sub-figures (a)-(g). Please correct this mis-labelling.
2. For metal nanoparticles, surface plasmon resonance is usually the cause of photothermal effect. Please comment in the main text if the photothermal effect of In/Al₂O₃ was caused by plasmonic localized heating or other mechanisms. The authors may refer to the following literature on those mechanisms.
Chem Catalysis, 2(1), 2022, pp 52-83, <https://doi.org/10.1016/j.checat.2021.10.005>
Research, vol. 2021, Article ID 979432, 2021, <https://doi.org/10.34133/2021/9794329>
3. In Figure 5a, the reactant and intermediate adsorption energies in In-In₂O₃ are lower than those in In₂O₃. But it is not clear what are those adsorbate binding configurations on surfaces. The authors should provide these structures in figures in the supporting information.
4. For In-In₂O₃, did the authors compare the adsorption energies of adsorbate (e.g., CO₂ and H) on the In and In₂O₃ portions of the surface structure, respectively? The authors need to rationalize or clarify if the catalytic active sites are on the In, In₂O₃, or both surfaces for the In-In₂O₃ catalyst.

Dear Reviewers,

We appreciate you sincerely for your constructive comments and suggestions related to our manuscript. These comments and suggestions are very valuable for our present and further study, especially which greatly help us to improve our manuscript.

We have carefully thought of your questions and responded them point by point as following.

Manuscript ID: NCOMMS-21-49904

“Subsurface Oxygen Defects Electronically Interacting with Active Sites on In_2O_3 for Enhanced Photothermocatalytic CO_2 Reduction”

REVIEWER COMMENTS

Reviewer #1 (Remarks to the Author):

In this manuscript, photothermocatalytic reduction of CO_2 into CO with high selectivity was achieved on In_2O_3 catalyst. More importantly, the catalytic activity for CO_2 -to-CO conversion was dramatically increased by embedding metallic In on In_2O_3 . The synergy between subsurface oxygen defects and metallic In on the surface of In_2O_3 was proved to be responsible for the improved catalytic CO_2 reduction performance. This study is interesting. It is recommended to be published on Nature Communications if the following issues can be properly addressed.

(1) What I most concerned is the stability of the catalyst. Based on the Methods part, the In-Em In_2O_3 sample was prepared by treating In_2O_3 in H_2/Ar (1/9) atmosphere, while the CO_2 reduction reaction was carried out in an atmosphere containing a H_2 with a much higher concentration ($\text{H}_2/\text{CO}_2/\text{Ar}=9:3:8$) at 300°C . Based on the Supplementary Fig. 2, surface reduction of In_2O_3 can occur at over 200°C , thus, it is reasonable to speculate that the reduction of In-Em In_2O_3 by H_2 should proceed in catalytic CO_2 reduction process. This might produce much more O defects (or metallic In) and change the structure of the catalysts. Furthermore, it is known that metallic In has a low melting point that is below 200°C . Does the melting of the embedded In occur under the photothermocatalytic reaction conditions? If this happens, does it affect the catalytic activity of In-Em In_2O_3 ? Therefore, the time dependent catalytic activity of In-Em In_2O_3 should be performed in a much long time in order to prove the good stability of the catalyst (the time dependent activity was only performed for 30 min as shown in Supplementary Fig. 17). After this long time reaction, the structure, composition of the catalyst should be identified.

Reply: Thank you for your valuable comment. As for the first question, In-Em

In_2O_3 cannot be further reduced to produce more oxygen defects and metallic In during the photothermocatalysis. The reasons are listed as following: First, the reductive temperature of the preparation of In-Em In_2O_3 was 450°C , while the photothermocatalytic temperature for CO_2 reduction was 300°C . The lower reaction temperature is not enough for the further reduction of In-Em In_2O_3 . Related with the concentration of H_2 , the influence of temperature is much greater. Second, H_2 -TPR pattern does not show the surface reduction peak for In-Em In_2O_3 (Fig. R1), suggesting that the surface reduction degree has reached maximum. Furthermore, in CO_2 -TPD patterns (Fig. R2), it can be observed that the concentrations of surface oxygen defects remain unchanged after the photothermocatalytic reaction. The number of active sites for absorbing and activating CO_2 is still rare, implying that the photothermocatalytic process does not result in more oxygen defects. Moreover, XRD patterns display that the amount of metallic In of In-Em In_2O_3 does not increase after the reaction (Fig. R3). Though the amount of metallic In was decreased due to the oxidation of OH intermediate, XPS patterns (Fig. R4) demonstrate that the total content of oxygen defects is nearly unchanged. Therefore, In-Em In_2O_3 cannot be reduced to generate more oxygen defects and metallic In during the photothermocatalysis.

Fig. R1 H_2 -TPR patterns of In_2O_3 and In-Em In_2O_3 .

Fig. R2 CO_2 -TPR patterns of In-Em In_2O_3 and In-Em In_2O_3 -spent(1) where “1” refers to that the catalyst underwent one run of reaction.

Fig. R3 XRD patterns of In-Em In_2O_3 and In-Em In_2O_3 -spent(10) where “10” refers to that the catalyst underwent ten runs of reaction.

Fig. R4 O_{1s} XPS spectra of In-Em In_2O_3 and In-Em In_2O_3 -spent(10) where “10” refers to that the catalyst underwent ten runs of reaction.

For the second question, at the temperature of 300°C , the embedded metallic In would be melt. Here, metallic In is not the active sites of CO_2 reduction, only serving as the carrier for conducting electrons. As we know, melt metals also have a powerful capability of conducting electrons. Compared with the ordered metallic In, the electron conduction of the melt counterpart may be decreased. However, the metallic In is embedded into the In_2O_3 lattice and furthermore there is a very strong interaction between In_2O_3 and metallic In, thus the melting does not result in the aggregate and removal of metallic In.

For the third question, we have measured the time dependent catalytic activity over In-Em In_2O_3 to verify the stability of the catalyst. Under the same condition (300°C , $\text{H}_2/\text{CO}_2/\text{Ar} = 9:3:8$), the catalytic reaction over In-Em In_2O_3 was cycled for ten runs (the reaction time of each run was 1 hour). The result showed that the catalytic cycling performance over In-Em In_2O_3 is stable (Fig. R5). Then the structure, composition of the catalyst spent was identified to explain the nature of catalytic stability. First, as demonstrated in XRD patterns (Fig. R3), after the photothermocatalytic reaction, the content of metallic In of In-Em In_2O_3 was decreased because of the oxidation of the metallic In by the OH intermediate from the

dissociation of COOH. Both the embedded and supported metallic In would be oxidized, however, this change does not affect the activity of the catalyst. Second, XPS spectra were used to analyze the surface component of In-Em In₂O₃-spent. The atom percent of oxygen defects remains (Fig. R4). Moreover, the content of In⁰ on the surface of the catalyst spent was slightly decreased by 6.3% (Fig. R6). TPD patterns were used to analyze the amount of the outermost layer of surface oxygen defects. The results were summarized in Table R1. At only one run of reaction, the amount of CO₂ adsorbed was unchanged, indicating the similar content of surface oxygen defects, which can be used to explain the unchanged performance. After ten runs of reaction, though the amount of surface oxygen defects increases by a factor of 40 fold, it is much lower related to In₂O₃ (0.335 mmol g⁻¹), suggesting that the higher reactivity of such oxygen defects is still dominant. That seems that the content and reactivity of oxygen defects complement each other for the stable performance. Raman spectra demonstrate the same ratio of $\nu(\text{InOIn})$ vs. $\delta(\text{InO}_6)$ (Fig. R7), indicating the unchanged interaction mode between In₂O₃ and metallic In and similar concentration of oxygen defect complex (*O-In-(O)Vo-In-In* structure).

We would like to supplement the related discussion in the third paragraph of the “TOF activity over In-Em In₂O₃ is 866 times higher than that over In₂O₃ under light irradiation” part and the content in Supplementary Fig. 19 (including XRD patterns, O_{1s} XPS spectra, In_{3d} XPS spectra, Raman spectra of In-Em In₂O₃ and In-Em In₂O₃-spent) as below:

In Manuscript: “We have measured the time dependent catalytic activity over In-Em In₂O₃ to verify the stability of the catalyst. Under the same condition (300°C, H₂/CO₂/Ar= 9:3:8), the catalytic reaction over In-Em In₂O₃ was cycled for ten runs (the reaction time of each run was 1 hour). The result showed that the catalytic cycling performance over In-Em In₂O₃ is stable (Fig. 3b). The content, structure and composition of oxygen defects of the spent catalyst were revealed by the characterization measurement for the nature of performance stability (Supplementary Fig. 19, Supplementary Table 5).”

In Supplementary Fig. 19 as Supplementary Discussion: “The structure, composition of the catalyst spent was identified to explain the nature of catalytic stability. First, as demonstrated in XRD patterns (Supplementary Fig. 19a), after the photothermocatalytic reaction, the content of metallic In of In-Em In₂O₃ was decreased because of the oxidation of the metallic In by the OH intermediate formed via the dissociation of COOH. Both the embedded and supported metallic In would be oxidized, however, this change can not affect the activity of the catalyst. Second, XPS spectra were used to analyze the surface component of In-Em In₂O₃-spent. The atom percent of oxygen defects remains (Supplementary Fig. 19b). Moreover, the content of In⁰ on the surface of the catalyst spent was slightly decreased by 6.3% (Supplementary Fig. 19c). Therefore, it is most likely to deduce that the structure and content of oxygen defects on the (sub)surface were retained during the photothermocatalytic reaction. TPD patterns were used to analyze the amount of the outermost layer of surface oxygen defects. The results were summarized in Supplementary Table 5. At one run of reaction, the amount of CO₂ adsorbed was

unchanged, indicating the similar content of surface oxygen defects, which can be used to explain the unchanged performance. After ten runs of reaction, though the amount of surface oxygen defects increases by a factor of 40 fold, it is much lower related to In_2O_3 ($0.335 \text{ mmol g}^{-1}$), suggesting that the higher reactivity of such oxygen defects is still dominant. That seems that the content and reactivity of oxygen defects complement each other for stable performance. Raman spectra demonstrate the same ratio of $\nu(\text{InOIn})$ vs. $\delta(\text{InO}_6)$ (Supplementary Fig. 19d), indicating the unchanged interaction mode between In_2O_3 and metallic In and similar concentration of oxygen defect complex (O-In-(O)Vo-In-In structure).”

Fig. R5 Cycle experiment over In-Em In_2O_3 for 10 runs.

Fig. R6 In_{3d} XPS spectra of In-Em In_2O_3 and In-Em In_2O_3 -spent(10) where “10” refers to that the catalyst underwent ten runs of reaction.

Table R1 CO_2 amount adsorbed measured from CO_2 -TPD patterns.

Catalyst	In-Em In_2O_3	In-Em In_2O_3 -spent (1)	In-Em In_2O_3 -spent (10)
Adsorbed amount of CO_2 (mmol g^{-1})	~0.0015	~0.0018	0.062

Fig. R7 Raman spectra of In-Em In_2O_3 and In-Em In_2O_3 -spent(10) where “10” refers to that the catalyst underwent ten runs of reaction.

(2) Based on Supplementary Fig. 21, it can be seen that the CO_2 reduction activity of the In-Em In_2O_3 catalyst should be decided by temperature rising effect induced by the incident light, and the photocatalysis plays little effect on the CO_2 reduction especially in the visible and infrared regions. On the other hand, the electron delocalization of In-Em In_2O_3 was significantly influenced by the light illumination at the same reaction temperature as proved by the ESR in Supplementary Fig. 23 (It seems that this figure is not consistent with expression of the manuscript. As indicated in the text, the ESR signal was tested in the dark and under light irradiation, respectively. However, the light and dark signals were not marked in this figure or identified in figure caption). Based on my understanding, the contributions from thermal effect and photocatalysis were not clearly discussed and identified in the whole manuscript. I suggest that a clear reaction pathway should be added in the end of the manuscript, in which the roles of heat, photoinduced charges, and H_2 in CO_2 reduction should be clearly shown.

Reply: Thank you for your suggestion. Sorry for that the dark ESR signal was left out before and we have supplemented the original dark ESR signal in Supplementary Fig. 27. For the expression in-consistence involving the photocatalytic performance of CO_2 reduction and ESR signals, the explanation is as following: The photothermocatalytic type in our study belongs to light-induced thermocatalysis, wherein thermocatalysis plays a dominant role whereas photocatalytic effect makes minor contribution. The experiments and the literature reported suggest that only oxygen defects can adsorb and activate CO_2 but indeed this does not happen in the region without oxygen defects. Under light irradiation, In_2O_3 phase can generate a few photogenerated electrons. The photogenerated electrons would move to the surface of In_2O_3 phase. However, few photogenerated electrons can be delivered to CO_2 adsorbed because the active region on the catalyst surface is too small. Only a few active sites which adsorb CO_2 are distributed at the interface between metallic In and In_2O_3 phase as verified by CO_2 -TPD (Fig. 2b), while no active sites are on the surface of In_2O_3 phase. Therefore, though irradiation to In_2O_3 generates more photogenerated charges, the photothermocatalytic performance does not increase significantly.

We would like to supplement the related discussion in the end of the manuscript as below:

“It is worth noting that most of the photogenerated charges in In_2O_3 portion are not delivered to CO_2 adsorbed specifically at the oxygen defects on the surface because of very small active region in spite of increasing ESR signals upon light irradiation as demonstrated in Supplementary Fig. 27.”

Finally, a clear reaction pathway has been added in the end of the manuscript, in which the roles of heat, photoinduced charges, and H_2 in CO_2 reduction are clearly shown.

The roles of heat: The light-induced heat is the dominant role in the photothermocatalysis, facilitating the transfer of charge carriers, excited vibration of the related species and the phonon of the ground state of In_2O_3 , thus lowering the reaction barrier of CO_2 reduction.

The roles of photogenerated charge: The photocatalytic effect makes minor contribution to the photothermocatalysis. There are two types of directions for the photogenerated charge: One is generating heat through electron-hole recombination while the other is delivering into CO_2 for CO_2 activation but this has little effect because most of the photogenerated charges cannot be transferred to the few active sites which adsorb CO_2 .

The roles of H_2 : H atoms dissociated from H_2 mainly participate in the formation of COOH and H_2O .

We would like to supplement the related discussion in the end of the manuscript as below:

“The reaction pathway in the system is clarified including the roles of heat, photogenerated carrier and H_2 in CO_2 reduction. For the present photothermocatalysis, the main contribution comes from the thermochemical pathway generated by light irradiation while the photochemical pathway makes minor contribution (Supplementary Fig. 25). Therefore, the catalytic reaction is called light-induced thermocatalysis⁴⁸. Here, a population of photons are absorbed by metallic In portion and oxygen defects (via “trap-assisted recombination”) and converted into thermal energy, respectively. The thermal chemistry facilitates the transfer of charge carrier, the excited vibration of the related species and the formation of the phonon of the ground state of In_2O_3 , which lower the reaction barrier of CO_2 reduction. Simultaneously, some photons are absorbed by In_2O_3 portion, forming photogenerated carrier. There are two types of evolution directions for the photogenerated carriers. One way is electron-hole recombination generating more heat, which is dominant, and the other is to be transferred to the adsorbed CO_2 . It is worth noting that most of the photogenerated charges in In_2O_3 portion are not delivered to CO_2 adsorbed specifically at the oxygen defects on the surface because of very small chemical reaction region in spite of increasing ESR signals upon light irradiation as demonstrated in Supplementary Fig. 27. If the photogenerated electrons can interact with CO_2 adsorbed, the energetic electrons would be injected, generating anion species⁵⁸. Otherwise, the photochemical pathway makes minor contribution to the

catalysis. The H_2 is dissociated into two H species in either a heterolytic or a homolytic way with the participation of lattice In and O^{59-62} , which binds with CO_2 to form COOH or binds with OH from dissociation of COOH to form H_2O , respectively⁷. However, the atmosphere including H_2 would not increment surface oxygen defects of In_2O_3 and metallic In during catalytic process, as indicated by H_2 -TPR pattern (Supplementary Fig. 2) and XRD pattern (Supplementary Fig. 19a).”

(3) Based on the O 1s XPS spectra of In_2O_3 and In-Em In_2O_3 , similar O defects were contained in the two samples. However, the colour and light absorption properties of the two samples are dramatically different. This phenomenon should be explained.

Reply: Thank you for your valuable comment. The light absorption property of In_2O_3 mainly originates from the electronic transition from valence band to conduction band. The band gap of In_2O_3 is ca. 3.0 eV primarily for UV-light absorption. The light absorption property of In-Em In_2O_3 include In_2O_3 semiconductor and metallic In. The former mainly leads to UV-light absorption while the latter causes the full-spectrum absorption. Therefore, though the content of oxygen defects is similar, the metallic In makes In-Em In_2O_3 exhibit dramatically different colour and light absorption property.

We would like to supplement the related discussion at the last sentence of the first paragraph of the “Subsurface oxygen defects surrounding metallic In” part as below:

“But their color and light absorption properties are dramatically different, because the light absorption characteristics of metallic In endows In-Em In_2O_3 full-spectrum absorption (referring to the last paragraph in the performance part).”

(4) Based on the In XPS spectra in Supplementary Fig. 10, only the shift of the peaks was observed after the formation of metallic In in In_2O_3 . Based on my understanding, there are two types of In (metallic and oxidation states) in the In-Em In_2O_3 sample. This phenomenon should be explained.

Reply: Thanks for this important suggestion. The In_{3d} XPS spectra are deconvoluted into two types of In species, including In^{3+} and In^0 . Their atom percents are 20.3% and 79.7%, respectively (Fig. R8).

We would like to supplement the related discussion at the first paragraph of the “Subsurface oxygen defects surrounding metallic In” part as below:

“Through peak deconvolution to In-Em In_2O_3 , two groups of characteristic splitting peaks are attributed to In^{3+} (444.7 and 452.3 eV, atom percent: 20.3%) and In^0 (444.1 and 451.7 eV, atom percent: 79.7%), respectively (Supplementary Fig. 11). The former belongs to In_2O_3 phase whereas the latter aggregates to form metallic In in the In_2O_3 lattice.”

Fig. R8 In_{3d} XPS spectra of In₂O₃ and In-Em In₂O₃.

(5) The reaction time for catalytic CO₂ reduction should be clearly indicated for calculating TOF and the mass and area specific activity.

Reply: According to your suggestion, we have indicated the reaction time for catalytic CO₂ reduction for calculating TOF and the mass and area specific activity.

We would like to supplement the related information at the “TOF activity over In-Em In₂O₃ is 866 times higher than that over In₂O₃ under light irradiation” part and experimental section as below:

“The reaction time of CO₂ reduction for calculating TOF and the mass and area specific activity as following is 1 hour unless special reaction time is mentioned.”

(6) For studying the overall morphology of the sample, SEM images of In₂O₃ and In-Em In₂O₃ should be shown.

Reply: Thank you for your suggestion. We have provided the SEM images of In₂O₃ and In-Em In₂O₃ (Fig. R9).

We would like to supplement the related information at the first paragraph of the “Temperature-dependent surface reconstruction of In₂O₃ nanoflakes” part as below:

“Field emission scanning electron microscope (FE-SEM, Supplementary Fig. 5) demonstrates the two-dimensional irregular overall morphologies of In₂O₃ and In-Em In₂O₃.”

Fig. R9 SEM images of (a) In_2O_3 and (b) In-Em In_2O_3 .

(7) In order to objectively understand the activity of the catalyst, the CO_2 reduction activity of In-Em In_2O_3 should be also compared with the recent works on the CO_2 reduction by thermal catalysis.

Reply: Thank you for your valuable comment. We have investigated a lot of related references and sorted out the table of performance comparison including more than 40 catalysts, as shown in Supplementary Table 4, which further indicates that our catalyst exhibits the most superior TOF activity of CO production.

We would like to supplement the table and references in Supplementary Table 4 as below:

Supplementary Table 4 Performance comparison of different catalysts for CO_2 hydrogenation into CO.

Catalyst	Reaction temp. ($^{\circ}\text{C}$)	CO_2 / H_2	TOF (h^{-1})	Catalytic type	Ref.
In-Em In_2O_3	300, 380	1/3	2990, 7615	Photothermal catalysis	This work
In_2O_3 NPs	330	1/3	17	Photothermal catalysis	-
In- In_2O_3 NPs	360	1/3	197	Photothermal catalysis	-
Black $\text{In}_2\text{O}_{3-x}$	370	1/1	1084 ^a	Photothermal catalysis	18

Black $\text{In}_2\text{O}_{3-x}$	300	1/1	1152 ^b	Photothermal catalysis	19
$\text{In}_2\text{O}_{3-x}(\text{OH})_y$	150	1/1	0.001 ^a	Photothermal catalysis	20
c- In_2O_3	350	1/2	1764 ^c	Thermal catalysis	21
h- In_2O_3	350	1/2	1638 ^c	Thermal catalysis	21
$\text{In}_2\text{O}_{3-x}(\text{OH})_y$ superstructure	R. T.	1/1	0.17 ^a	Photocatalysis	22
$\text{Bi}_2\text{O}_{3-x}$	200	1/2	280 ^a	Photothermal catalysis	23
(X)Ni- In_2O_3 (X: 1, 5, 10, 15)	280	1/4	10, 14, 17, 41 ^a	Thermal catalysis	24
$\text{In}_2\text{O}_3/\text{ZrO}_2$	250	1/3	0.84 ^a	Thermal catalysis	25
$\text{In}_2\text{O}_3/\text{CeO}_2\text{-h}$	250	1/3	22 ^a	Thermal catalysis	25
2Y8In/ ZrO_2	300	1/4	0.71 ^a	Thermal catalysis	26
3La10In/ ZrO_2	300	1/4	0.92 ^a	Thermal catalysis	26
Ru/ TiO_2	200	3/1	14 ^d	Thermal catalysis	27
Ni/ SiO_2	350	1/4	32 ^d	Thermal catalysis	28
Pd/ ZnO	250	1/3	1 ^d	Thermal catalysis	29
Ni/HY (molecular sieve)	300	1/4	157 ^d	Thermal catalysis	30
PtCo/ TiO_2	300	1/2	2093 ^d	Thermal catalysis	31
[PPN] [$\text{RuCl}_3(\text{CO})_3$]	160	1/3	17~19	Thermal catalysis	32
Co ZrO_x	340	1/4	265 ^a	Thermal catalysis	33
Ru/ SiO_2	300	1/1	342 ^d	Thermal catalysis	34
Pt ₁ / TiO_2	250	1/4	1296 ^d	Thermal catalysis	35
Pd/ CeO_2	275	1/3	200~500 ^d	Thermal catalysis	36
Rh/ Fe_3O_4	300	1/4	468 ^d , 312 ^a	Thermal catalysis	37

ZrO ₂ @Pd/Si O ₂	450	1/3	3137 ^d	Thermal catalysis	38
Ru/CeO ₂	330	1/4	1008 ^d	Thermal catalysis	39

^a The value in the literature is equal to CO production rate/number of active sites where number of active sites was obtained from CO₂-TPD tests. ^b The value originates from the literature wherein number of active sites was estimated from XPS spectra. ^c The value is estimated afterward wherein the number of active sites was obtained according to the CO₂-TPD tests in the literature. ^d The active sites are reported to be metal catalyst dispersed on oxide and the number is measured via FT-IR and H₂-TPR.

New references in the Supplementary Information as below:

24. Frei, M. S. et al. Nanostructure of nickel-promoted indium oxide catalysts drives selectivity in CO₂ hydrogenation. *Nat. Commun.* **12**, 1960 (2021).
25. Regalado Vera, C. Y. et al. Mechanistic understanding of support effect on the activity and selectivity of indium oxide catalysts for CO₂ hydrogenation. *Chem. Eng. J.* **426**, 131764 (2021).
26. Chou, C. Y., Lobo, R. F. Direct conversion of CO₂ into methanol over promoted indium oxide-based catalysts. *Appl. Catal., A* **583**, 117144 (2019).
27. Matsubu, J. C.; Yang, V. N.; Christopher, P. Isolated metal active site concentration and stability control catalytic CO₂ reduction selectivity. *J. Am. Chem. Soc.* **137**, 3076–3084 (2015).
28. Aldana, P. A. U. et al. Catalytic CO₂ valorization into CH₄ on Ni-based ceria-zirconia. reaction mechanism by operando IR spectroscopy. *Catal. Today* **215**, 201–207 (2013).
29. Bahruji, H. et al. Pd/ZnO catalysts for direct CO₂ hydrogenation to methanol. *J. Catal.* **343**, 133–146 (2016).
30. Aziz, M. A. A. et al. Highly active Ni-promoted mesostructured silica nanoparticles for CO₂ methanation. *Appl. Catal., B* **147**, 359–368 (2014).
31. Kattel, S. et al. CO₂ Hydrogenation over oxide-supported PtCo catalysts: The role of the oxide support in determining the product selectivity. *Angew. Chem. Int. Ed.* **55**, 7968–7973 (2016).
32. Tsuchiya, K.; Huang, J. D.; Tominaga, K. I. Reverse water-gas shift reaction catalyzed by mononuclear Ru complexes. *ACS Catal.* **3**, 2865–2868 (2013).
33. Dostagir, N. H. M. et al. Co single atoms in ZrO₂ with inherent oxygen vacancies for selective hydrogenation of CO₂ to CO. *ACS Catal.* **11**, 9450–9461 (2021).
34. Mansour, H.; Iglesia, E. Mechanistic connections between CO₂ and CO hydrogenation on dispersed ruthenium nanoparticles. *J. Am. Chem. Soc.* **143**, 11582–11594 (2021).
35. Chen, L. et al. Unlocking the catalytic potential of TiO₂-supported Pt single atoms for the reverse water-gas shift reaction by altering their chemical environment. *J. Am. Chem. Soc. (Au)* **1**, 977–986 (2021).
36. Cao, F. et al. Size-controlled synthesis of Pd nanocatalysts on defect-engineered

- CeO₂ for CO₂ hydrogenation. *ACS Appl. Mater. Inter.* **13**, 24957–24965 (2021).
37. Zhu, Y. et al. Environment of metal-O-Fe bonds enabling high activity in CO₂ reduction on single metal atoms and on supported nanoparticles. *J. Am. Chem. Soc.* **143**, 5540–5549 (2021).
38. Du, Y. P. et al. Engineering the ZrO₂-Pd interface for selective CO₂ hydrogenation by overcoating an atomically dispersed Pd precatalyst. *ACS Catal.* **10**, 12058–12070 (2020).
39. Wang, Y. et al. Site-selective CO₂ reduction over highly dispersed Ru-SnO_x sites derived from a [Ru@Sn₉]₆-zintl cluster. *ACS Catal.* **10**, 7808–7819 (2020).

Reviewer #2 (Remarks to the Author):

The work by Wei et al., examined the use of Indium Oxide-based catalysts for the photo-thermal catalytic conversion of carbon dioxide to carbon monoxide. By controlling reduction, the work examined the role of ‘subsurface’ oxygen vacancies in driving selective conversion. Whilst the findings are interesting and the differences in performances are notable, there are distinct holes in understanding and some contradictions in the characterization which I believe undermine the conclusions drawn by the work. As such, I cannot recommend the work for publication.

1) The main flaw in the work is the manner in which comparisons are drawn with respect to performance of the two materials. The use of TOF as a basis for catalyst comparison, is acceptable when the active sites are well defined, for example when they exist as metal deposits on supports. In this case, the active sites were defined by using CO₂ TPDs whereby the amount desorbed was used to estimate molar quantity of active sites. Whilst oxygen vacancies are evidenced in the literature as the active sites, the CO₂ TPDs are not an accurate measure of active sites under reaction conditions. The reaction occurs in a reducing environment, with heat and light. All of these can (and likely will) lead to in situ reduction and increase in the surface active sites. Using this basis is therefore highly inaccurate and unreliable. In fact, it is even stated in the manuscript that these active sites changes (Supp. Fig. 18)

Reply: Thank you for your valuable comment. We have also done a lot of thinking and experiments to rationalize that the content and structure of the oxygen defects of In-Em In₂O₃ remain unchanged during the catalytic process. Four key points to support that are mentioned as following:

A review of the literature shows that for all the catalysts containing oxygen defects, the oxygen defects may change to some extent. However, accounting for the limitations of experimental techniques and experimental conditions, most of the relevant literature for the analysis of oxygen defects are based on the oxygen defects before the reaction, which hardly affects the accuracy of the results. CO₂-TPD technique has been applied in many documents to detect the influence of oxygen

defect on catalysis and measure the amount of oxygen defects, such as **Example 1st** (From Gao P. et al, *Nat. Chem.*, **2017**, 1019, DOI:10.1038/nchem.2794), **Example 2nd** (From Lei F. et al, *J. Am. Chem. Soc.*, **2014**, 6826, DOI: 10.1021/ja501866r) and **Example 3st**. (From Linsebigler, A. L. et al, *Chem. Rev.* **1995**, 95, 735-758). The related examples are too many, not limited to In_2O_3 . Simultaneously, it suggests that the change of oxygen defects is relative, not absolute and we can ensure that the content and structure of oxygen defects remain nearly only if we control suitable preparation and reaction conditions. For the investigation of dynamic process of oxygen defects, it is a very challenging and thorny issue, which requires too much more work to make it clear. We believe that our study can contribute to it together with many excellent studies.

Also, we have tried to control the reaction conditions to retain the structure and chemical environment of oxygen defects as far as possible, such as low-temperature catalysis and short-time catalysis (Supplementary Fig. 18). The results also verify our conclusion. The oxygen species generated would not annihilate oxygen defects if the oxygen species can be eliminated in time. In our study, besides H species which would combine with O, metallic In could play that role during the photothermocatalysis but it needs to be exploited in the following work.

One important point to know is that the H_2 reductive temperature of the preparation of In-Em In_2O_3 was 450°C , while the photothermocatalytic temperature for CO_2 reduction was only 300°C . The lower reaction temperature is not enough for the further reduction of In-Em In_2O_3 . Related with the concentration of H_2 , the influence of temperature is much greater. Moreover, H_2 -TPR pattern does not show the surface reduction peak for In-Em In_2O_3 (Fig. R1), suggesting that the surface reduction degree has reached maximum, that is, without generation of other more oxygen defects.

Fig. R1 H_2 -TPR patterns of In_2O_3 and In-Em In_2O_3 .

Finally, we have measured the time dependent catalytic activity of In-Em In_2O_3 to verify the stability of the catalyst. Under the same condition (300°C , $\text{H}_2/\text{CO}_2/\text{Ar}=9:3:8$), the catalytic reaction over In-Em In_2O_3 was cycled for ten runs (the reaction time of each run was 1 hour). The result showed that the catalytic cycling performance over In-Em In_2O_3 is stable (Fig. R5), indicating that the structure and content of oxygen defects are nearly unchanged during the photothermocatalysis. We

also measured XPS, CO₂-TPD and Raman spectra of the catalysts after the reaction. All this suggests that the content and chemical structure of oxygen defects of In-Em In₂O₃ remain nearly unchanged.

Fig. R4 O_{1s} XPS spectra of In-Em In₂O₃ and In-Em In₂O₃-spent(10) where “10” refers to that the catalyst underwent ten runs of reaction.

Fig. R5 Cycle experiment over In-Em In₂O₃ for 10 runs.

Fig. R6 In_{3d} XPS spectra of In-Em In₂O₃ and In-Em In₂O₃-spent(10) where “10” refers to that the catalyst underwent ten runs of reaction.

Fig. R7 Raman spectra of In-Em In_2O_3 and In-Em In_2O_3 -spent(10) where “10” refers to that the catalyst underwent ten runs of reaction.

2) The surface/subsurface reduction at 400°C is stated by literature support however the measured TPDs do not indicate this. Further, TPRs of the reduced and pre-treated samples should be provided for comparison and to further understand the reduction of the materials.

Reply: We would like to further explain the TPD measurement. The CO_2 -TPD measurement (Fig. 2b) has indicated the subsurface reduction and approved the existence of subsurface oxygen defects. The metallic In embedded in In_2O_3 lattice was etched by diluted HCl solution and thus some subsurface oxygen defects were exposed, leading to an obvious chemisorption peak resulting from subsurface oxygen defects (as exhibited by the purple curve of Fig. 2b). The temperature of 400°C mentioned in H_2 -TPR patterns (in fact 450°C) refer to its onset temperature for bulk reduction of In_2O_3 . Besides, we have provided H_2 -TPR patterns (Fig. R1) of In_2O_3 and In-Em In_2O_3 for comparison. The result suggests that the surface of In-Em In_2O_3 have already reached maximum reduction degree.

We would like to supplement the related discussion at the first paragraph of the “Temperature-dependent surface reconstruction of In_2O_3 nanoflakes” part as below:

“It is worth noting that the further reduction to In-Em In_2O_3 does not cause any surface reduction as shown in Supplementary Fig. 2 because the surface reduction degree of In-Em In_2O_3 has reached maximum.”

Fig. R1 H_2 -TPR patterns of In_2O_3 and In-Em In_2O_3 .

3) How can the presence of surface oxygen vacancies be ruled out? These could be playing a key role along with the increased light-to-heat conversion as a function of having black coloured reduced In present.

Reply: We agree with your opinion that the oxygen defects can increase light-to-heat conversion via “trap-assisted recombination”. However, for In-Em In₂O₃ in our study, metallic In plays a dominant role in photothermal effect compared with oxygen defects. The content of surface oxygen defects of In₂O₃ is far higher than that of In-Em In₂O₃, but seen from Supplementary Fig. 17a, b, the temperature-ramp rate of In-Em In₂O₃ is faster than that of In₂O₃. Therefore, the light-to-heat conversion from oxygen defects is lower than that from metallic In.

We would like to supplement the related discussion in the last paragraph of the “TOF activity over In-Em In₂O₃ is 866 times higher than that over In₂O₃ under light irradiation” part and in the end of the manuscript as below:

“Photothermal conversion over the system is one of the key factors dictating the photothermocatalytic performance of CO₂ reduction. The photothermal conversion capability of In₂O₃ is rather low (Supplementary Fig. 17a) because of the light absorption of the wavelength below 500 nm and dominant radiative emission. Compared with In₂O₃, In-Em In₂O₃ displayed a more efficient photothermal conversion due to full-spectral light absorption (to near-infrared light) and a high probability of nonradiative relaxation (Supplementary Fig. 17). The photothermal effect of In-Em In₂O₃ originates from oxygen defects and light absorption characteristics of metallic In. As reported, oxygen defects can create mid-gap energy state and thus increase light-to-heat conversion due to enhanced light absorption⁷ and “trap-assisted recombination”⁴⁶. Compared with oxygen defects, metallic In of In-Em In₂O₃ plays a dominant role in light-to-heat conversion which heats up the metal lattice by electron-phonon scattering. Due to superior thermal conduction of metallic In, the concentrated energy in metallic In is then rapidly transferred to the active site of In₂O₃ portion for CO₂ reduction via phonon-phonon relaxation⁴⁷. However, in the future, it is worth investigating which of the light absorption modes from metallic In exhibit the highest efficiency of light-to-heat conversion, including interband-transition absorption, intraband-transition absorption, plasmon-resonance absorption⁴⁶⁻⁴⁸.”

4) The evidence of the subsurface CO₂ is unclear, only AFM is used to confirm this. Perhaps a mild passivation to oxidize the surface then measure would be helpful.

Reply: Thank you for your suggestion. We adopted 3% H₂O₂ solution as a mild oxidant to treat In-Em In₂O₃ and then tested its photothermocatalytic performance. The original motivate was to passivate the oxygen defects on the surface of In-Em In₂O₃ with diluted hydrogen peroxide solution, and to observe whether the subsurface oxygen defects interact with CO₂. However, hydrogen peroxide would oxide the surface of metallic In and increment the content of oxygen defects around metallic In. The performance was increased by ~7% compared with In-Em In₂O₃ (Fig. R10). The XRD, XPS, Raman and CO₂-TPD were measured to explain the phenomenon.

It can be seen from XRD patterns that the diffraction peak intensity ratio of metallic In/In₂O₃ does not change (Fig. R11), indicating that the hydrogen peroxide treatment does not change the overall ratio of metallic In/In₂O₃. Moreover, In_{3d} XPS spectra show that the surface component of In species change very slightly (Fig. R12). Also, the interaction mode between In₂O₃ and metallic In and concentration of oxygen defect complex (*O-In-(O)Vo-In-In* structure) remain as indicated by Raman spectra (Fig. R13). It can be surmised that a very small part of metallic In on the surface was oxidized, forming a very thin layer of indium oxide which possesses a certain amount of surface oxygen defects around metallic In. Indeed, the O_{1s} XPS spectra show that the content of surface oxygen defects was increased by 6.7% (Fig. R14). TPD patterns verify that the increased oxygen defects are on the outermost surface (Fig. R15). This further confirms our accuracy of our conclusion that the oxygen defects around metallic In exhibit the stronger reactivity.

We would like to supplement the related discussion in the sixth paragraph of the “TOF activity over In-Em In₂O₃ is 866 times higher than that over In₂O₃ under light irradiation” part and Supplementary Fig. 22 as below:

In Manuscript: “In addition, we adopted mild hydrogen peroxide to oxidize the surface of metallic In of In-Em In₂O₃ (named as In-Em In₂O₃(H₂O₂)), forming a thin layer of In₂O₃ with more oxygen defects. This can create more surface oxygen defects around metallic In. The performance evaluation displays the enhanced activity over In-Em In₂O₃(H₂O₂) compared with In-Em In₂O₃ (Supplementary Fig. 21). The characterizations (Supplementary Fig. 22) suggest that the overall structure of In-Em In₂O₃(H₂O₂) changes little, but the content of surface oxygen defects around increases. This further verifies that the oxygen defects around metallic In exhibit the higher reactivity.”

In Supplementary Fig. 22 as Supplementary Discussion: “It can be seen from XRD patterns (Supplementary Fig. 22a) that the diffraction peak intensity ratio of metallic In/In₂O₃ does not change, indicating that the hydrogen peroxide treatment does not change the overall ratio of metallic In/In₂O₃. Moreover, In_{3d} XPS spectra show that the surface component of In species change slightly (Supplementary Fig. 22b). Also, the interaction mode between In₂O₃ and metallic In and concentration of oxygen defect complex (*O-In-(O)Vo-In-In* structure) remain as indicated by Raman spectra (Supplementary Fig. 22c). It can be surmised that the metallic In on the surface was oxidized, forming a very thin layer of indium oxide which possesses a certain amount of surface oxygen defects. Indeed, the O_{1s} XPS spectra show that the content of surface oxygen defects was increased by 6.7% (Supplementary Fig. 22d). CO₂-TPD patterns verify that the increased oxygen defects are on the outermost surface (Supplementary Fig. 22e). This further confirms the accuracy of our conclusion that the oxygen defects around metallic In exhibit the higher reactivity.”

Fig. R10 Performance comparison over In-Em In₂O₃ and In-Em In₂O₃(H₂O₂).

Fig. R11 XRD patterns of In-Em In₂O₃ and In-Em In₂O₃(H₂O₂).

Fig. R12 In_{3d} XPS spectra of In-Em In₂O₃ and In-Em In₂O₃(H₂O₂).

Fig. R13 Raman spectra of In-Em In_2O_3 and In-Em $\text{In}_2\text{O}_3(\text{H}_2\text{O}_2)$.

Fig. R14 O_{1s} XPS spectra of In-Em In_2O_3 and In-Em $\text{In}_2\text{O}_3(\text{H}_2\text{O}_2)$.

Fig. R15 CO_2 -TPD patterns of In-Em $\text{In}_2\text{O}_3(\text{H}_2\text{O}_2)$.

5) The conversion of carbon dioxide is small $\ll 25\%$, with such small conversions, improvements in performance of 2 fold are small, particularly without error bars. In this case, error bars are essential to draw real conclusions.

Reply: Thank you for your valuable comment. The error bars in Fig. 3, Supplementary Fig. 16, 22 and 35 were added. One point to know is that mechanism investigation involving in one single active site is our study subject, rather than apparent CO production. From Supplementary Table 4, the In_2O_3 -based catalyst with

metallic In embedded surpasses many catalysts reported with respect to the reactivity of one active site. The TOF activity of In-Em In_2O_3 demonstrates the very obvious distinction related with other catalysts.

Reviewer #3 (Remarks to the Author):

In this manuscript, the authors reported that subsurface oxygen defects in the In-embedded In_2O_3 catalyst is much more active (by 866 or 376 folds) than the In_2O_3 catalyst, for reductive conversion of CO_2 to CO under photothermal or thermal conditions. The authors characterized the In/ In_2O_3 catalyst using XRD, HRTEM, XPS, TPD, ESR, PAS, FTIR, and XANES. The subsurface oxygen defects were studied by combining the results of XPS, and PAS, and XANES. The electron transfer from subsurface oxygen defect was observed by ESR under light irradiation. By DFT calculations, the authors found that the electron is delocalized from the subsurface oxygen defects onto the embedded In nanoparticles, resulting in the enhanced binding of CO_2 , H, and intermediates and thus significantly improved CO_2 conversion. This work is surely of great interests to the catalytic conversion of CO_2 , using subsurface oxygen defects in designed catalysts. Comprehensive experimental and computational analyses were performed to support the main argument. I would suggest the manuscript to be published in the journal of Nature Communications, subjected to the following minor revision.

1. In Figure 1 caption, the descriptions of (a)-(f) do not match with sub-figures (a)-(g). Please correct this mis-labelling.

Reply: Thank you for your careful reviewing. We have corrected the mis-labelling. Please see the caption of Fig. 1. Moreover, we also have carefully checked our manuscript for correcting similar mis-spelling or mis-labelling which has been revised and marked by red color in the manuscript.

2. For metal nanoparticles, surface plasmon resonance is usually the cause of photothermal effect. Please comment in the main text if the photothermal effect of In/ In_2O_3 was caused by plasmonic localized heating or other mechanisms. The authors may refer to the following literature on those mechanisms.

Chem Catalysis, 2(1), 2022, pp 52-83, <https://doi.org/10.1016/j.checat.2021.10.005>
Research, vol. 2021, Article ID 979432, 2021, <https://doi.org/10.34133/2021/9794329>

Reply: Thank you for your valuable comment and literature recommendation. We have carefully read the two literature and the papers deepen our understanding of photothermocatalysis. Some metals such as Au, Ag and Cu possess specific light absorption for plasmon resonance. Such absorption is relevant with the size of metal. The plasmon-resonance absorption of metallic In has not been reported. Here, the

light-to-heat conversion of In-Em In_2O_3 is mainly realized by metallic In as shown in Supplementary Fig. 17. However, it is not certain which of the light absorption modes from metallic In exhibit the highest efficiency of light-to-heat conversion, including interband-transition absorption, intraband-transition absorption, plasmon-resonance absorption.

We would like to supplement the related discussion in the last paragraph of the “TOF activity over In-Em In_2O_3 is 866 times higher than that over In_2O_3 under light irradiation” part as below:

“Photothermal conversion over the system is one of the key factors dictating the photothermocatalytic performance of CO_2 reduction. The photothermal conversion capability of In_2O_3 is rather low (Supplementary Fig. 17a) because of the light absorption of the wavelength below 500 nm and dominant radiative emission. Compared with In_2O_3 , In-Em In_2O_3 displayed a more efficient photothermal conversion due to full-spectral light absorption (to near-infrared light) and a high probability of nonradiative relaxation (Supplementary Fig. 17). The photothermal effect of In-Em In_2O_3 originates from oxygen defects and light absorption characteristics of metallic In. As reported, oxygen defects can create mid-gap energy state and thus increase light-to-heat conversion due to enhanced light absorption⁷ and “trap-assisted recombination”⁴⁶. Compared with oxygen defects, metallic In of In-Em In_2O_3 plays a dominant role in light-to-heat conversion which heats up the metal lattice by electron-phonon scattering. Due to superior thermal conduction of metallic In, the concentrated energy in metallic In is then rapidly transferred to the active site of In_2O_3 portion for CO_2 reduction via phonon-phonon relaxation⁴⁷. However, in the future, it is worth investigating which of the light absorption modes from metallic In exhibit the highest efficiency of light-to-heat conversion, including interband-transition absorption, intraband-transition absorption, plasmon-resonance absorption⁴⁶⁻⁴⁸.”

3. In Figure 5a, the reactant and intermediate adsorption energies in In- In_2O_3 are lower than those in In_2O_3 . But it is not clear what are those adsorbate binding configurations on surfaces. The authors should provide these structures in figures in the supporting information.

Reply: Thank you for your valuable comment. We have supplemented the related figures and made a discussion. The figures are in Supplementary Fig. 39 and 40. The relevant discussions were added in page 45 and 46 of Supplementary information.

Fig. R15 The adsorbate binding configurations on the surface of In₂O₃. (a) Adsorbed CO₂ and H. (b) COOH intermediate. (c) CO and OH from COOH dissociation.

“The C atom and one of the O atoms of CO₂ are bound with O and In of In₂O₃ around oxygen vacancy, respectively. The two C-O bonds of CO₂ form a bond angle of 120°, suggesting CO₂ activation. The bent structure of CO₂ is the feature of CO₂ anion¹⁶. With the collision coupling of H and the other O atom of CO₂, COOH forms accompanied by the O end linking with In atom. After the dissociation of COOH, the OH is bridged with two In atoms.”

Fig. R16 The adsorbate binding configurations on the surface of In-In₂O₃. (a) Adsorbed CO₂ and H. (b) COOH intermediate. (c) CO and OH from COOH dissociation.

“The adsorption of H and CO₂ is on the In₂O₃ side and the interaction mode is similar with the result above on In₂O₃. H and CO₂ collide at the interface to form COOH. One of the O atoms of COOH is connected with In⁰ atom. Finally, COOH is dissociated accompanied by the OH bridged with two In atoms.”

4. For In-In₂O₃, did the authors compare the adsorption energies of adsorbate (e.g., CO₂ and H) on the In and In₂O₃ portions of the surface structure, respectively? The authors need to rationalize or clarify if the catalytic active sites are on the In, In₂O₃, or both surfaces for the In-In₂O₃ catalyst.

Reply: Thank you for your suggestion. We have theoretically calculated the adsorption energies of adsorbates (e.g., CO₂ and H) on metallic In and In₂O₃, respectively. The comparison of adsorption energies is in the table as following:

Catalyst	In ₂ O ₃	Metallic In	In-In ₂ O ₃
CO ₂	0.287 eV	-0.081 eV	-0.750 eV
H	0.235 eV	-1.058 eV	-3.292 eV

Through DFT calculations, the adsorption energies of CO₂ and H on metallic In are lower than that on In₂O₃, but the adsorption energies on In-In₂O₃ (namely at the interface between metallic In and In₂O₃) are much lower compared with metallic In and In₂O₃. This in theory indicates that the active sites are neither on the metallic In nor on the In₂O₃, but most likely at the interface between metallic In and In₂O₃. Moreover, we have employed the sole metallic In as the photothermal catalyst for CO₂ reduction, which showed that the metallic In does not have the capability for CO₂ reduction. This experimentally excludes metallic In as the active sites. Accordingly, the active sites are not the In₂O₃ and metallic In but rather the oxygen defects at their interface.

We would like to supplement the related discussion in the “Enhanced CO₂ adsorption and activation” part as below:

“The adsorption energies of CO₂ and H on metallic In were calculated to be -0.081 and -1.058 eV, respectively. The adsorption energies of CO₂ and H on In-In₂O₃ (namely at the interface between metallic In and In₂O₃) are much lower than that on In₂O₃ and metallic In, which most likely evidences that the active sites are at the interface between metallic In and In₂O₃. This is consistent with the conclusion above that the oxygen defects at the interface function as the active sites for CO₂ reduction.”

Other revisions for editors:

1. Correct the note of “see supplementary information”.
2. We have read the formatting instructions carefully and revised some incorrect formats, e.g. the word number in the abstract (less than 150), the panel style in the figures and the placement position of ACKNOWLEDGEMENTS, AUTHOR CONTRIBUTIONS and COMPETING INTERESTS.
3. We have added a “Data Availability” section after the Methods section but before the References.

That’s all. Thank you again for discussing about our study.

Yours sincerely,

OUYANG Shuxin, PhD, Professor

Key Laboratory of Pesticide and Chemical Biology of Ministry of Education

College of Chemistry,

Central China Normal University,

Wuhan 430079, China

E-mail: oysx@mail.ccnu.edu.cn

REVIEWERS' COMMENTS

Reviewer #1 (Remarks to the Author):

Thanks for the responses and revisions of the authors, which are satisfactory. Now, I recommend this manuscript for publication in Nature Communications.

Reviewer #2 (Remarks to the Author):

The authors have considered the feedback and done extensive further work and characterisation to support their findings. I am happy to recommend this work for publication.

Reviewer #3 (Remarks to the Author):

The authors have properly addressed all my questions and comments. The overall manuscript is in good quality. I would recommend the manuscript to be published in the journal of Nature Communications.

Dear Reviewers,

We appreciate you sincerely for your constructive comments and suggestions related to our manuscript. These comments and suggestions are very valuable for our present and further study, especially which greatly help us to improve our manuscript.

We have carefully thought of your questions and responded them point by point as following.

Manuscript ID: NCOMMS-21-49904

“Subsurface Oxygen Defects Electronically Interacting with Active Sites on In_2O_3 for Enhanced Photothermocatalytic CO_2 Reduction”

REVIEWER COMMENTS

Reviewer #1 (Remarks to the Author):

In this manuscript, photothermocatalytic reduction of CO_2 into CO with high selectivity was achieved on In_2O_3 catalyst. More importantly, the catalytic activity for CO_2 -to-CO conversion was dramatically increased by embedding metallic In on In_2O_3 . The synergy between subsurface oxygen defects and metallic In on the surface of In_2O_3 was proved to be responsible for the improved catalytic CO_2 reduction performance. This study is interesting. It is recommended to be published on Nature Communications if the following issues can be properly addressed.

(1) What I most concerned is the stability of the catalyst. Based on the Methods part, the In-Em In_2O_3 sample was prepared by treating In_2O_3 in H_2/Ar (1/9) atmosphere, while the CO_2 reduction reaction was carried out in an atmosphere containing a H_2 with a much higher concentration ($\text{H}_2/\text{CO}_2/\text{Ar}=9:3:8$) at 300°C . Based on the Supplementary Fig. 2, surface reduction of In_2O_3 can occur at over 200°C , thus, it is reasonable to speculate that the reduction of In-Em In_2O_3 by H_2 should proceed in catalytic CO_2 reduction process. This might produce much more O defects (or metallic In) and change the structure of the catalysts. Furthermore, it is known that metallic In has a low melting point that is below 200°C . Does the melting of the embedded In occur under the photothermocatalytic reaction conditions? If this happens, does it affect the catalytic activity of In-Em In_2O_3 ? Therefore, the time dependent catalytic activity of In-Em In_2O_3 should be performed in a much long time in order to prove the good stability of the catalyst (the time dependent activity was only performed for 30 min as shown in Supplementary Fig. 17). After this long time reaction, the structure, composition of the catalyst should be identified.

Reply: Thank you for your valuable comment. As for the first question, In-Em

In_2O_3 cannot be further reduced to produce more oxygen defects and metallic In during the photothermocatalysis. The reasons are listed as following: First, the reductive temperature of the preparation of In-Em In_2O_3 was 450°C , while the photothermocatalytic temperature for CO_2 reduction was 300°C . The lower reaction temperature is not enough for the further reduction of In-Em In_2O_3 . Related with the concentration of H_2 , the influence of temperature is much greater. Second, H_2 -TPR pattern does not show the surface reduction peak for In-Em In_2O_3 (Fig. R1), suggesting that the surface reduction degree has reached maximum. Furthermore, in CO_2 -TPD patterns (Fig. R2), it can be observed that the concentrations of surface oxygen defects remain unchanged after the photothermocatalytic reaction. The number of active sites for absorbing and activating CO_2 is still rare, implying that the photothermocatalytic process does not result in more oxygen defects. Moreover, XRD patterns display that the amount of metallic In of In-Em In_2O_3 does not increase after the reaction (Fig. R3). Though the amount of metallic In was decreased due to the oxidation of OH intermediate, XPS patterns (Fig. R4) demonstrate that the total content of oxygen defects is nearly unchanged. Therefore, In-Em In_2O_3 cannot be reduced to generate more oxygen defects and metallic In during the photothermocatalysis.

Fig. R1 H_2 -TPR patterns of In_2O_3 and In-Em In_2O_3 .

Fig. R2 CO_2 -TPR patterns of In-Em In_2O_3 and In-Em In_2O_3 -spent(1) where “1” refers to that the catalyst underwent one run of reaction.

Fig. R3 XRD patterns of In-Em In_2O_3 and In-Em In_2O_3 -spent(10) where “10” refers to that the catalyst underwent ten runs of reaction.

Fig. R4 O_{1s} XPS spectra of In-Em In_2O_3 and In-Em In_2O_3 -spent(10) where “10” refers to that the catalyst underwent ten runs of reaction.

For the second question, at the temperature of 300°C , the embedded metallic In would be melt. Here, metallic In is not the active sites of CO_2 reduction, only serving as the carrier for conducting electrons. As we know, melt metals also have a powerful capability of conducting electrons. Compared with the ordered metallic In, the electron conduction of the melt counterpart may be decreased. However, the metallic In is embedded into the In_2O_3 lattice and furthermore there is a very strong interaction between In_2O_3 and metallic In, thus the melting does not result in the aggregate and removal of metallic In.

For the third question, we have measured the time dependent catalytic activity over In-Em In_2O_3 to verify the stability of the catalyst. Under the same condition (300°C , $\text{H}_2/\text{CO}_2/\text{Ar}= 9:3:8$), the catalytic reaction over In-Em In_2O_3 was cycled for ten runs (the reaction time of each run was 1 hour). The result showed that the catalytic cycling performance over In-Em In_2O_3 is stable (Fig. R5). Then the structure, composition of the catalyst spent was identified to explain the nature of catalytic stability. First, as demonstrated in XRD patterns (Fig. R3), after the photothermocatalytic reaction, the content of metallic In of In-Em In_2O_3 was decreased because of the oxidation of the metallic In by the OH intermediate from the

dissociation of COOH. Both the embedded and supported metallic In would be oxidized, however, this change does not affect the activity of the catalyst. Second, XPS spectra were used to analyze the surface component of In-Em In₂O₃-spent. The atom percent of oxygen defects remains (Fig. R4). Moreover, the content of In⁰ on the surface of the catalyst spent was slightly decreased by 6.3% (Fig. R6). TPD patterns were used to analyze the amount of the outermost layer of surface oxygen defects. The results were summarized in Table R1. At only one run of reaction, the amount of CO₂ adsorbed was unchanged, indicating the similar content of surface oxygen defects, which can be used to explain the unchanged performance. After ten runs of reaction, though the amount of surface oxygen defects increases by a factor of 40 fold, it is much lower related to In₂O₃ (0.335 mmol g⁻¹), suggesting that the higher reactivity of such oxygen defects is still dominant. That seems that the content and reactivity of oxygen defects complement each other for the stable performance. Raman spectra demonstrate the same ratio of $\nu(\text{InOIn})$ vs. $\delta(\text{InO}_6)$ (Fig. R7), indicating the unchanged interaction mode between In₂O₃ and metallic In and similar concentration of oxygen defect complex (*O-In-(O)Vo-In-In* structure).

We would like to supplement the related discussion in the third paragraph of the “TOF activity over In-Em In₂O₃ is 866 times higher than that over In₂O₃ under light irradiation” part and the content in Supplementary Fig. 19 (including XRD patterns, O_{1s} XPS spectra, In_{3d} XPS spectra, Raman spectra of In-Em In₂O₃ and In-Em In₂O₃-spent) as below:

In Manuscript: “We have measured the time dependent catalytic activity over In-Em In₂O₃ to verify the stability of the catalyst. Under the same condition (300°C, H₂/CO₂/Ar= 9:3:8), the catalytic reaction over In-Em In₂O₃ was cycled for ten runs (the reaction time of each run was 1 hour). The result showed that the catalytic cycling performance over In-Em In₂O₃ is stable (Fig. 3b). The content, structure and composition of oxygen defects of the spent catalyst were revealed by the characterization measurement for the nature of performance stability (Supplementary Fig. 19, Supplementary Table 5).”

In Supplementary Fig. 19 as Supplementary Discussion: “The structure, composition of the catalyst spent was identified to explain the nature of catalytic stability. First, as demonstrated in XRD patterns (Supplementary Fig. 19a), after the photothermocatalytic reaction, the content of metallic In of In-Em In₂O₃ was decreased because of the oxidation of the metallic In by the OH intermediate formed via the dissociation of COOH. Both the embedded and supported metallic In would be oxidized, however, this change can not affect the activity of the catalyst. Second, XPS spectra were used to analyze the surface component of In-Em In₂O₃-spent. The atom percent of oxygen defects remains (Supplementary Fig. 19b). Moreover, the content of In⁰ on the surface of the catalyst spent was slightly decreased by 6.3% (Supplementary Fig. 19c). Therefore, it is most likely to deduce that the structure and content of oxygen defects on the (sub)surface were retained during the photothermocatalytic reaction. TPD patterns were used to analyze the amount of the outermost layer of surface oxygen defects. The results were summarized in Supplementary Table 5. At one run of reaction, the amount of CO₂ adsorbed was

unchanged, indicating the similar content of surface oxygen defects, which can be used to explain the unchanged performance. After ten runs of reaction, though the amount of surface oxygen defects increases by a factor of 40 fold, it is much lower related to In_2O_3 ($0.335 \text{ mmol g}^{-1}$), suggesting that the higher reactivity of such oxygen defects is still dominant. That seems that the content and reactivity of oxygen defects complement each other for stable performance. Raman spectra demonstrate the same ratio of $\nu(\text{InOIn})$ vs. $\delta(\text{InO}_6)$ (Supplementary Fig. 19d), indicating the unchanged interaction mode between In_2O_3 and metallic In and similar concentration of oxygen defect complex (O-In-(O)Vo-In-In structure).”

Fig. R5 Cycle experiment over In-Em In_2O_3 for 10 runs.

Fig. R6 In_{3d} XPS spectra of In-Em In_2O_3 and In-Em In_2O_3 -spent(10) where “10” refers to that the catalyst underwent ten runs of reaction.

Table R1 CO_2 amount adsorbed measured from CO_2 -TPD patterns.

Catalyst	In-Em In_2O_3	In-Em In_2O_3 -spent (1)	In-Em In_2O_3 -spent (10)
Adsorbed amount of CO_2 (mmol g^{-1})	~0.0015	~0.0018	0.062

Fig. R7 Raman spectra of In-Em In_2O_3 and In-Em In_2O_3 -spent(10) where “10” refers to that the catalyst underwent ten runs of reaction.

(2) Based on Supplementary Fig. 21, it can be seen that the CO_2 reduction activity of the In-Em In_2O_3 catalyst should be decided by temperature rising effect induced by the incident light, and the photocatalysis plays little effect on the CO_2 reduction especially in the visible and infrared regions. On the other hand, the electron delocalization of In-Em In_2O_3 was significantly influenced by the light illumination at the same reaction temperature as proved by the ESR in Supplementary Fig. 23 (It seems that this figure is not consistent with expression of the manuscript. As indicated in the text, the ESR signal was tested in the dark and under light irradiation, respectively. However, the light and dark signals were not marked in this figure or identified in figure caption). Based on my understanding, the contributions from thermal effect and photocatalysis were not clearly discussed and identified in the whole manuscript. I suggest that a clear reaction pathway should be added in the end of the manuscript, in which the roles of heat, photoinduced charges, and H_2 in CO_2 reduction should be clearly shown.

Reply: Thank you for your suggestion. Sorry for that the dark ESR signal was left out before and we have supplemented the original dark ESR signal in Supplementary Fig. 27. For the expression in-consistence involving the photocatalytic performance of CO_2 reduction and ESR signals, the explanation is as following: The photothermocatalytic type in our study belongs to light-induced thermocatalysis, wherein thermocatalysis plays a dominant role whereas photocatalytic effect makes minor contribution. The experiments and the literature reported suggest that only oxygen defects can adsorb and activate CO_2 but indeed this does not happen in the region without oxygen defects. Under light irradiation, In_2O_3 phase can generate a few photogenerated electrons. The photogenerated electrons would move to the surface of In_2O_3 phase. However, few photogenerated electrons can be delivered to CO_2 adsorbed because the active region on the catalyst surface is too small. Only a few active sites which adsorb CO_2 are distributed at the interface between metallic In and In_2O_3 phase as verified by CO_2 -TPD (Fig. 2b), while no active sites are on the surface of In_2O_3 phase. Therefore, though irradiation to In_2O_3 generates more photogenerated charges, the photothermocatalytic performance does not increase significantly.

We would like to supplement the related discussion in the end of the manuscript as below:

“It is worth noting that most of the photogenerated charges in In_2O_3 portion are not delivered to CO_2 adsorbed specifically at the oxygen defects on the surface because of very small active region in spite of increasing ESR signals upon light irradiation as demonstrated in Supplementary Fig. 27.”

Finally, a clear reaction pathway has been added in the end of the manuscript, in which the roles of heat, photoinduced charges, and H_2 in CO_2 reduction are clearly shown.

The roles of heat: The light-induced heat is the dominant role in the photothermocatalysis, facilitating the transfer of charge carriers, excited vibration of the related species and the phonon of the ground state of In_2O_3 , thus lowering the reaction barrier of CO_2 reduction.

The roles of photogenerated charge: The photocatalytic effect makes minor contribution to the photothermocatalysis. There are two types of directions for the photogenerated charge: One is generating heat through electron-hole recombination while the other is delivering into CO_2 for CO_2 activation but this has little effect because most of the photogenerated charges cannot be transferred to the few active sites which adsorb CO_2 .

The roles of H_2 : H atoms dissociated from H_2 mainly participate in the formation of COOH and H_2O .

We would like to supplement the related discussion in the end of the manuscript as below:

“The reaction pathway in the system is clarified including the roles of heat, photogenerated carrier and H_2 in CO_2 reduction. For the present photothermocatalysis, the main contribution comes from the thermochemical pathway generated by light irradiation while the photochemical pathway makes minor contribution (Supplementary Fig. 25). Therefore, the catalytic reaction is called light-induced thermocatalysis⁴⁸. Here, a population of photons are absorbed by metallic In portion and oxygen defects (via “trap-assisted recombination”) and converted into thermal energy, respectively. The thermal chemistry facilitates the transfer of charge carrier, the excited vibration of the related species and the formation of the phonon of the ground state of In_2O_3 , which lower the reaction barrier of CO_2 reduction. Simultaneously, some photons are absorbed by In_2O_3 portion, forming photogenerated carrier. There are two types of evolution directions for the photogenerated carrier. One way is electron-hole recombination generating more heat, which is dominant, and the other is to be transferred to CO_2 adsorbed. It is worth noting that most of the photogenerated charges in In_2O_3 portion are not delivered to CO_2 adsorbed specifically at the oxygen defects on the surface because of very small chemical reaction region in spite of increasing ESR signals upon light irradiation as demonstrated in Supplementary Fig. 27. If the photogenerated electrons can interact with CO_2 adsorbed, the energetic electrons would be injected, generating anion species⁵⁸. Otherwise, the photochemical pathway makes minor contribution to the

catalysis. The H_2 is dissociated into two H species in either a heterolytic or a homolytic way with the participation of lattice In and O^{59-62} , which binds with CO_2 to form COOH or binds with OH from dissociation of COOH to form H_2O , respectively⁷. However, the atmosphere including H_2 would not increment surface oxygen defects of In_2O_3 and metallic In during catalytic process, as indicated by H_2 -TPR pattern (Supplementary Fig. 2) and XRD pattern (Supplementary Fig. 19a).”

(3) Based on the O 1s XPS spectra of In_2O_3 and In-Em In_2O_3 , similar O defects were contained in the two samples. However, the colour and light absorption properties of the two samples are dramatically different. This phenomenon should be explained.

Reply: Thank you for your valuable comment. The light absorption property of In_2O_3 mainly originates from the electronic transition from valence band to conduction band. The band gap of In_2O_3 is ca. 3.0 eV primarily for UV-light absorption. The light absorption property of In-Em In_2O_3 include In_2O_3 semiconductor and metallic In. The former mainly leads to UV-light absorption while the latter causes the full-spectrum absorption. Therefore, though the content of oxygen defects is similar, the metallic In makes In-Em In_2O_3 exhibit dramatically different colour and light absorption property.

We would like to supplement the related discussion at the last sentence of the first paragraph of the “Subsurface oxygen defects surrounding metallic In” part as below:

“But their color and light absorption properties are dramatically different, because the light absorption characteristics of metallic In endows In-Em In_2O_3 full-spectrum absorption (referring to the last paragraph in the performance part).”

(4) Based on the In XPS spectra in Supplementary Fig. 10, only the shift of the peaks was observed after the formation of metallic In in In_2O_3 . Based on my understanding, there are two types of In (metallic and oxidation states) in the In-Em In_2O_3 sample. This phenomenon should be explained.

Reply: Thanks for this important suggestion. The In_{3d} XPS spectra are deconvoluted into two types of In species, including In^{3+} and In^0 . Their atom percents are 20.3% and 79.7%, respectively (Fig. R8).

We would like to supplement the related discussion at the first paragraph of the “Subsurface oxygen defects surrounding metallic In” part as below:

“Through peak deconvolution to In-Em In_2O_3 , two groups of characteristic splitting peaks are attributed to In^{3+} (444.7 and 452.3 eV, atom percent: 20.3%) and In^0 (444.1 and 451.7 eV, atom percent: 79.7%), respectively (Supplementary Fig. 11). The former belongs to In_2O_3 phase whereas the latter aggregates to form metallic In in the In_2O_3 lattice.”

Fig. R8 In_{3d} XPS spectra of In₂O₃ and In-Em In₂O₃.

(5) The reaction time for catalytic CO₂ reduction should be clearly indicated for calculating TOF and the mass and area specific activity.

Reply: According to your suggestion, we have indicated the reaction time for catalytic CO₂ reduction for calculating TOF and the mass and area specific activity.

We would like to supplement the related information at the “TOF activity over In-Em In₂O₃ is 866 times higher than that over In₂O₃ under light irradiation” part and experimental section as below:

“The reaction time of CO₂ reduction for calculating TOF and the mass and area specific activity as following is 1 hour unless special reaction time is mentioned.”

(6) For studying the overall morphology of the sample, SEM images of In₂O₃ and In-Em In₂O₃ should be shown.

Reply: Thank you for your suggestion. We have provided the SEM images of In₂O₃ and In-Em In₂O₃ (Fig. R9).

We would like to supplement the related information at the first paragraph of the “Temperature-dependent surface reconstruction of In₂O₃ nanoflakes” part as below:

“Field emission scanning electron microscope (FE-SEM, Supplementary Fig. 5) demonstrates the two-dimensional irregular overall morphologies of In₂O₃ and In-Em In₂O₃.”

Fig. R9 SEM images of (a) In_2O_3 and (b) In-Em In_2O_3 .

(7) In order to objectively understand the activity of the catalyst, the CO_2 reduction activity of In-Em In_2O_3 should be also compared with the recent works on the CO_2 reduction by thermal catalysis.

Reply: Thank you for your valuable comment. We have investigated a lot of related references and sorted out the table of performance comparison including more than 40 catalysts, as shown in Supplementary Table 4, which further indicates that our catalyst exhibits the most superior TOF activity of CO production.

We would like to supplement the table and references in Supplementary Table 4 as below:

Supplementary Table 4 Performance comparison of different catalysts for CO_2 hydrogenation into CO.

Catalyst	Reaction temp. ($^{\circ}\text{C}$)	CO_2 / H_2	TOF (h^{-1})	Catalytic type	Ref.
In-Em In_2O_3	300, 380	1/3	2990, 7615	Photothermal catalysis	This work
In_2O_3 NPs	330	1/3	17	Photothermal catalysis	-
In- In_2O_3 NPs	360	1/3	197	Photothermal catalysis	-
Black $\text{In}_2\text{O}_{3-x}$	370	1/1	1084 ^a	Photothermal catalysis	18

Black $\text{In}_2\text{O}_{3-x}$	300	1/1	1152 ^b	Photothermal catalysis	19
$\text{In}_2\text{O}_{3-x}(\text{OH})_y$	150	1/1	0.001 ^a	Photothermal catalysis	20
c- In_2O_3	350	1/2	1764 ^c	Thermal catalysis	21
h- In_2O_3	350	1/2	1638 ^c	Thermal catalysis	21
$\text{In}_2\text{O}_{3-x}(\text{OH})_y$ superstructure	R. T.	1/1	0.17 ^a	Photocatalysis	22
$\text{Bi}_2\text{O}_{3-x}$	200	1/2	280 ^a	Photothermal catalysis	23
(X)Ni- In_2O_3 (X: 1, 5, 10, 15)	280	1/4	10, 14, 17, 41 ^a	Thermal catalysis	24
$\text{In}_2\text{O}_3/\text{ZrO}_2$	250	1/3	0.84 ^a	Thermal catalysis	25
$\text{In}_2\text{O}_3/\text{CeO}_2\text{-h}$	250	1/3	22 ^a	Thermal catalysis	25
2Y8In/ ZrO_2	300	1/4	0.71 ^a	Thermal catalysis	26
3La10In/ ZrO_2	300	1/4	0.92 ^a	Thermal catalysis	26
Ru/ TiO_2	200	3/1	14 ^d	Thermal catalysis	27
Ni/ SiO_2	350	1/4	32 ^d	Thermal catalysis	28
Pd/ ZnO	250	1/3	1 ^d	Thermal catalysis	29
Ni/HY (molecular sieve)	300	1/4	157 ^d	Thermal catalysis	30
PtCo/ TiO_2	300	1/2	2093 ^d	Thermal catalysis	31
[PPN] [$\text{RuCl}_3(\text{CO})_3$]	160	1/3	17~19	Thermal catalysis	32
Co ZrO_x	340	1/4	265 ^a	Thermal catalysis	33
Ru/ SiO_2	300	1/1	342 ^d	Thermal catalysis	34
Pt ₁ / TiO_2	250	1/4	1296 ^d	Thermal catalysis	35
Pd/ CeO_2	275	1/3	200~500 ^d	Thermal catalysis	36
Rh/ Fe_3O_4	300	1/4	468 ^d , 312 ^a	Thermal catalysis	37

ZrO ₂ @Pd/Si O ₂	450	1/3	3137 ^d	Thermal catalysis	38
Ru/CeO ₂	330	1/4	1008 ^d	Thermal catalysis	39

^a The value in the literature is equal to CO production rate/number of active sites where number of active sites was obtained from CO₂-TPD tests. ^b The value originates from the literature wherein number of active sites was estimated from XPS spectra. ^c The value is estimated afterward wherein the number of active sites was obtained according to the CO₂-TPD tests in the literature. ^d The active sites are reported to be metal catalyst dispersed on oxide and the number is measured via FT-IR and H₂-TPR.

New references in the Supplementary Information as below:

24. Frei, M. S. et al. Nanostructure of nickel-promoted indium oxide catalysts drives selectivity in CO₂ hydrogenation. *Nat. Commun.* **12**, 1960 (2021).
25. Regalado Vera, C. Y. et al. Mechanistic understanding of support effect on the activity and selectivity of indium oxide catalysts for CO₂ hydrogenation. *Chem. Eng. J.* **426**, 131764 (2021).
26. Chou, C. Y., Lobo, R. F. Direct conversion of CO₂ into methanol over promoted indium oxide-based catalysts. *Appl. Catal., A* **583**, 117144 (2019).
27. Matsubu, J. C.; Yang, V. N.; Christopher, P. Isolated metal active site concentration and stability control catalytic CO₂ reduction selectivity. *J. Am. Chem. Soc.* **137**, 3076–3084 (2015).
28. Aldana, P. A. U. et al. Catalytic CO₂ valorization into CH₄ on Ni-based ceria-zirconia. reaction mechanism by operando IR spectroscopy. *Catal. Today* **215**, 201–207 (2013).
29. Bahruji, H. et al. Pd/ZnO catalysts for direct CO₂ hydrogenation to methanol. *J. Catal.* **343**, 133–146 (2016).
30. Aziz, M. A. A. et al. Highly active Ni-promoted mesostructured silica nanoparticles for CO₂ methanation. *Appl. Catal., B* **147**, 359–368 (2014).
31. Kattel, S. et al. CO₂ Hydrogenation over oxide-supported PtCo catalysts: The role of the oxide support in determining the product selectivity. *Angew. Chem. Int. Ed.* **55**, 7968–7973 (2016).
32. Tsuchiya, K.; Huang, J. D.; Tominaga, K. I. Reverse water-gas shift reaction catalyzed by mononuclear Ru complexes. *ACS Catal.* **3**, 2865–2868 (2013).
33. Dostagir, N. H. M. et al. Co single atoms in ZrO₂ with inherent oxygen vacancies for selective hydrogenation of CO₂ to CO. *ACS Catal.* **11**, 9450–9461 (2021).
34. Mansour, H.; Iglesia, E. Mechanistic connections between CO₂ and CO hydrogenation on dispersed ruthenium nanoparticles. *J. Am. Chem. Soc.* **143**, 11582–11594 (2021).
35. Chen, L. et al. Unlocking the catalytic potential of TiO₂-supported Pt single atoms for the reverse water-gas shift reaction by altering their chemical environment. *J. Am. Chem. Soc. (Au)* **1**, 977–986 (2021).
36. Cao, F. et al. Size-controlled synthesis of Pd nanocatalysts on defect-engineered

- CeO₂ for CO₂ hydrogenation. *ACS Appl. Mater. Inter.* **13**, 24957–24965 (2021).
37. Zhu, Y. et al. Environment of metal-O-Fe bonds enabling high activity in CO₂ reduction on single metal atoms and on supported nanoparticles. *J. Am. Chem. Soc.* **143**, 5540–5549 (2021).
38. Du, Y. P. et al. Engineering the ZrO₂-Pd interface for selective CO₂ hydrogenation by overcoating an atomically dispersed Pd precatalyst. *ACS Catal.* **10**, 12058–12070 (2020).
39. Wang, Y. et al. Site-selective CO₂ reduction over highly dispersed Ru-SnO_x sites derived from a [Ru@Sn₉]₆-zintl cluster. *ACS Catal.* **10**, 7808–7819 (2020).

Reviewer #2 (Remarks to the Author):

The work by Wei et al., examined the use of Indium Oxide-based catalysts for the photo-thermal catalytic conversion of carbon dioxide to carbon monoxide. By controlling reduction, the work examined the role of ‘subsurface’ oxygen vacancies in driving selective conversion. Whilst the findings are interesting and the differences in performances are notable, there are distinct holes in understanding and some contradictions in the characterization which I believe undermine the conclusions drawn by the work. As such, I cannot recommend the work for publication.

1) The main flaw in the work is the manner in which comparisons are drawn with respect to performance of the two materials. The use of TOF as a basis for catalyst comparison, is acceptable when the active sites are well defined, for example when they exist as metal deposits on supports. In this case, the active sites were defined by using CO₂ TPDs whereby the amount desorbed was used to estimate molar quantity of active sites. Whilst oxygen vacancies are evidenced in the literature as the active sites, the CO₂ TPDs are not an accurate measure of active sites under reaction conditions. The reaction occurs in a reducing environment, with heat and light. All of these can (and likely will) lead to in situ reduction and increase in the surface active sites. Using this basis is therefore highly inaccurate and unreliable. In fact, it is even stated in the manuscript that these active sites changes (Supp. Fig. 18)

Reply: Thank you for your valuable comment. We have also done a lot of thinking and experiments to rationalize that the content and structure of the oxygen defects of In-Em In₂O₃ remain unchanged during the catalytic process. Four key points to support that are mentioned as following:

A review of the literature shows that for all the catalysts containing oxygen defects, the oxygen defects may change to some extent. However, accounting for the limitations of experimental techniques and experimental conditions, most of the relevant literature for the analysis of oxygen defects are based on the oxygen defects before the reaction, which hardly affects the accuracy of the results. CO₂-TPD technique has been applied in many documents to detect the influence of oxygen

defect on catalysis and measure the amount of oxygen defects, such as **Example 1st** (From Gao P. et al, *Nat. Chem.*, **2017**, 1019, DOI:10.1038/nchem.2794), **Example 2nd** (From Lei F. et al, *J. Am. Chem. Soc.*, **2014**, 6826, DOI: 10.1021/ja501866r) and **Example 3st**. (From Linsebigler, A. L. et al, *Chem. Rev.* **1995**, 95, 735-758). The related examples are too many, not limited to In_2O_3 . Simultaneously, it suggests that the change of oxygen defects is relative, not absolute and we can ensure that the content and structure of oxygen defects remain nearly only if we control suitable preparation and reaction conditions. For the investigation of dynamic process of oxygen defects, it is a very challenging and thorny issue, which requires too much more work to make it clear. We believe that our study can contribute to it together with many excellent studies.

Also, we have tried to control the reaction conditions to retain the structure and chemical environment of oxygen defects as far as possible, such as low-temperature catalysis and short-time catalysis (Supplementary Fig. 18). The results also verify our conclusion. The oxygen species generated would not annihilate oxygen defects if the oxygen species can be eliminated in time. In our study, besides H species which would combine with O, metallic In could play that role during the photothermocatalysis but it needs to be exploited in the following work.

One important point to know is that the H_2 reductive temperature of the preparation of In-Em In_2O_3 was 450°C , while the photothermocatalytic temperature for CO_2 reduction was only 300°C . The lower reaction temperature is not enough for the further reduction of In-Em In_2O_3 . Related with the concentration of H_2 , the influence of temperature is much greater. Moreover, H_2 -TPR pattern does not show the surface reduction peak for In-Em In_2O_3 (Fig. R1), suggesting that the surface reduction degree has reached maximum, that is, without generation of other more oxygen defects.

Fig. R1 H_2 -TPR patterns of In_2O_3 and In-Em In_2O_3 .

Finally, we have measured the time dependent catalytic activity of In-Em In_2O_3 to verify the stability of the catalyst. Under the same condition (300°C , $\text{H}_2/\text{CO}_2/\text{Ar}=9:3:8$), the catalytic reaction over In-Em In_2O_3 was cycled for ten runs (the reaction time of each run was 1 hour). The result showed that the catalytic cycling performance over In-Em In_2O_3 is stable (Fig. R5), indicating that the structure and content of oxygen defects are nearly unchanged during the photothermocatalysis. We

also measured XPS, CO₂-TPD and Raman spectra of the catalysts after the reaction. All this suggests that the content and chemical structure of oxygen defects of In-Em In₂O₃ remain nearly unchanged.

Fig. R4 O_{1s} XPS spectra of In-Em In₂O₃ and In-Em In₂O₃-spent(10) where “10” refers to that the catalyst underwent ten runs of reaction.

Fig. R5 Cycle experiment over In-Em In₂O₃ for 10 runs.

Fig. R6 In_{3d} XPS spectra of In-Em In₂O₃ and In-Em In₂O₃-spent(10) where “10” refers to that the catalyst underwent ten runs of reaction.

Fig. R7 Raman spectra of In-Em In_2O_3 and In-Em In_2O_3 -spent(10) where “10” refers to that the catalyst underwent ten runs of reaction.

2) The surface/subsurface reduction at 400°C is stated by literature support however the measured TPDs do not indicate this. Further, TPRs of the reduced and pre-treated samples should be provided for comparison and to further understand the reduction of the materials.

Reply: We would like to further explain the TPD measurement. The CO_2 -TPD measurement (Fig. 2b) has indicated the subsurface reduction and approved the existence of subsurface oxygen defects. The metallic In embedded in In_2O_3 lattice was etched by diluted HCl solution and thus some subsurface oxygen defects were exposed, leading to an obvious chemisorption peak resulting from subsurface oxygen defects (as exhibited by the purple curve of Fig. 2b). The temperature of 400°C mentioned in H_2 -TPR patterns (in fact 450°C) refer to its onset temperature for bulk reduction of In_2O_3 . Besides, we have provided H_2 -TPR patterns (Fig. R1) of In_2O_3 and In-Em In_2O_3 for comparison. The result suggests that the surface of In-Em In_2O_3 have already reached maximum reduction degree.

We would like to supplement the related discussion at the first paragraph of the “Temperature-dependent surface reconstruction of In_2O_3 nanoflakes” part as below:

“It is worth noting that the further reduction to In-Em In_2O_3 does not cause any surface reduction as shown in Supplementary Fig. 2 because the surface reduction degree of In-Em In_2O_3 has reached maximum.”

Fig. R1 H_2 -TPR patterns of In_2O_3 and In-Em In_2O_3 .

3) How can the presence of surface oxygen vacancies be ruled out? These could be playing a key role along with the increased light-to-heat conversion as a function of having black coloured reduced In present.

Reply: We agree with your opinion that the oxygen defects can increase light-to-heat conversion via “trap-assisted recombination”. However, for In-Em In₂O₃ in our study, metallic In plays a dominant role in photothermal effect compared with oxygen defects. The content of surface oxygen defects of In₂O₃ is far higher than that of In-Em In₂O₃, but seen from Supplementary Fig. 17a, b, the temperature-ramp rate of In-Em In₂O₃ is faster than that of In₂O₃. Therefore, the light-to-heat conversion from oxygen defects is lower than that from metallic In.

We would like to supplement the related discussion in the last paragraph of the “TOF activity over In-Em In₂O₃ is 866 times higher than that over In₂O₃ under light irradiation” part and in the end of the manuscript as below:

“Photothermal conversion over the system is one of the key factors dictating the photothermocatalytic performance of CO₂ reduction. The photothermal conversion capability of In₂O₃ is rather low (Supplementary Fig. 17a) because of the light absorption of the wavelength below 500 nm and dominant radiative emission. Compared with In₂O₃, In-Em In₂O₃ displayed a more efficient photothermal conversion due to full-spectral light absorption (to near-infrared light) and a high probability of nonradiative relaxation (Supplementary Fig. 17). The photothermal effect of In-Em In₂O₃ originates from oxygen defects and light absorption characteristics of metallic In. As reported, oxygen defects can create mid-gap energy state and thus increase light-to-heat conversion due to enhanced light absorption⁷ and “trap-assisted recombination”⁴⁶. Compared with oxygen defects, metallic In of In-Em In₂O₃ plays a dominant role in light-to-heat conversion which heats up the metal lattice by electron-phonon scattering. Due to superior thermal conduction of metallic In, the concentrated energy in metallic In is then rapidly transferred to the active site of In₂O₃ portion for CO₂ reduction via phonon-phonon relaxation⁴⁷. However, in the future, it is worth investigating which of the light absorption modes from metallic In exhibit the highest efficiency of light-to-heat conversion, including interband-transition absorption, intraband-transition absorption, plasmon-resonance absorption⁴⁶⁻⁴⁸.”

4) The evidence of the subsurface CO₂ is unclear, only AFM is used to confirm this. Perhaps a mild passivation to oxidize the surface then measure would be helpful.

Reply: Thank you for your suggestion. We adopted 3% H₂O₂ solution as a mild oxidant to treat In-Em In₂O₃ and then tested its photothermocatalytic performance. The original motivate was to passivate the oxygen defects on the surface of In-Em In₂O₃ with diluted hydrogen peroxide solution, and to observe whether the subsurface oxygen defects interact with CO₂. However, hydrogen peroxide would oxide the surface of metallic In and increment the content of oxygen defects around metallic In. The performance was increased by ~7% compared with In-Em In₂O₃ (Fig. R10). The XRD, XPS, Raman and CO₂-TPD were measured to explain the phenomenon.

It can be seen from XRD patterns that the diffraction peak intensity ratio of metallic In/In₂O₃ does not change (Fig. R11), indicating that the hydrogen peroxide treatment does not change the overall ratio of metallic In/In₂O₃. Moreover, In_{3d} XPS spectra show that the surface component of In species change very slightly (Fig. R12). Also, the interaction mode between In₂O₃ and metallic In and concentration of oxygen defect complex (*O-In-(O)Vo-In-In* structure) remain as indicated by Raman spectra (Fig. R13). It can be surmised that a very small part of metallic In on the surface was oxidized, forming a very thin layer of indium oxide which possesses a certain amount of surface oxygen defects around metallic In. Indeed, the O_{1s} XPS spectra show that the content of surface oxygen defects was increased by 6.7% (Fig. R14). TPD patterns verify that the increased oxygen defects are on the outermost surface (Fig. R15). This further confirms our accuracy of our conclusion that the oxygen defects around metallic In exhibit the stronger reactivity.

We would like to supplement the related discussion in the sixth paragraph of the “TOF activity over In-Em In₂O₃ is 866 times higher than that over In₂O₃ under light irradiation” part and Supplementary Fig. 22 as below:

In Manuscript: “In addition, we adopted mild hydrogen peroxide to oxidize the surface of metallic In of In-Em In₂O₃ (named as In-Em In₂O₃(H₂O₂)), forming a thin layer of In₂O₃ with more oxygen defects. This can create more surface oxygen defects around metallic In. The performance evaluation displays the enhanced activity over In-Em In₂O₃(H₂O₂) compared with In-Em In₂O₃ (Supplementary Fig. 21). The characterizations (Supplementary Fig. 22) suggest that the overall structure of In-Em In₂O₃(H₂O₂) changes little, but the content of surface oxygen defects around increases. This further verifies that the oxygen defects around metallic In exhibit the higher reactivity.”

In Supplementary Fig. 22 as Supplementary Discussion: “It can be seen from XRD patterns (Supplementary Fig. 22a) that the diffraction peak intensity ratio of metallic In/In₂O₃ does not change, indicating that the hydrogen peroxide treatment does not change the overall ratio of metallic In/In₂O₃. Moreover, In_{3d} XPS spectra show that the surface component of In species change slightly (Supplementary Fig. 22b). Also, the interaction mode between In₂O₃ and metallic In and concentration of oxygen defect complex (*O-In-(O)Vo-In-In* structure) remain as indicated by Raman spectra (Supplementary Fig. 22c). It can be surmised that the metallic In on the surface was oxidized, forming a very thin layer of indium oxide which possesses a certain amount of surface oxygen defects. Indeed, the O_{1s} XPS spectra show that the content of surface oxygen defects was increased by 6.7% (Supplementary Fig. 22d). CO₂-TPD patterns verify that the increased oxygen defects are on the outermost surface (Supplementary Fig. 22e). This further confirms the accuracy of our conclusion that the oxygen defects around metallic In exhibit the higher reactivity.”

Fig. R10 Performance comparison over In-Em In₂O₃ and In-Em In₂O₃(H₂O₂).

Fig. R11 XRD patterns of In-Em In₂O₃ and In-Em In₂O₃(H₂O₂).

Fig. R12 In_{3d} XPS spectra of In-Em In₂O₃ and In-Em In₂O₃(H₂O₂).

Fig. R13 Raman spectra of In-Em In_2O_3 and In-Em $\text{In}_2\text{O}_3(\text{H}_2\text{O}_2)$.

Fig. R14 O_{1s} XPS spectra of In-Em In_2O_3 and In-Em $\text{In}_2\text{O}_3(\text{H}_2\text{O}_2)$.

Fig. R15 CO_2 -TPD patterns of In-Em $\text{In}_2\text{O}_3(\text{H}_2\text{O}_2)$.

5) The conversion of carbon dioxide is small $\ll 25\%$, with such small conversions, improvements in performance of 2 fold are small, particularly without error bars. In this case, error bars are essential to draw real conclusions.

Reply: Thank you for your valuable comment. The error bars in Fig. 3, Supplementary Fig. 16, 22 and 35 were added. One point to know is that mechanism investigation involving in one single active site is our study subject, rather than apparent CO production. From Supplementary Table 4, the In_2O_3 -based catalyst with

metallic In embedded surpasses many catalysts reported with respect to the reactivity of one active site. The TOF activity of In-Em In_2O_3 demonstrates the very obvious distinction related with other catalysts.

Reviewer #3 (Remarks to the Author):

In this manuscript, the authors reported that subsurface oxygen defects in the In-embedded In_2O_3 catalyst is much more active (by 866 or 376 folds) than the In_2O_3 catalyst, for reductive conversion of CO_2 to CO under photothermal or thermal conditions. The authors characterized the In/ In_2O_3 catalyst using XRD, HRTEM, XPS, TPD, ESR, PAS, FTIR, and XANES. The subsurface oxygen defects were studied by combining the results of XPS, and PAS, and XANES. The electron transfer from subsurface oxygen defect was observed by ESR under light irradiation. By DFT calculations, the authors found that the electron is delocalized from the subsurface oxygen defects onto the embedded In nanoparticles, resulting in the enhanced binding of CO_2 , H, and intermediates and thus significantly improved CO_2 conversion. This work is surely of great interests to the catalytic conversion of CO_2 , using subsurface oxygen defects in designed catalysts. Comprehensive experimental and computational analyses were performed to support the main argument. I would suggest the manuscript to be published in the journal of Nature Communications, subjected to the following minor revision.

1. In Figure 1 caption, the descriptions of (a)-(f) do not match with sub-figures (a)-(g). Please correct this mis-labelling.

Reply: Thank you for your careful reviewing. We have corrected the mis-labelling. Please see the caption of Fig. 1. Moreover, we also have carefully checked our manuscript for correcting similar mis-spelling or mis-labelling which has been revised and marked by red color in the manuscript.

2. For metal nanoparticles, surface plasmon resonance is usually the cause of photothermal effect. Please comment in the main text if the photothermal effect of In/ In_2O_3 was caused by plasmonic localized heating or other mechanisms. The authors may refer to the following literature on those mechanisms.

Chem Catalysis, 2(1), 2022, pp 52-83, <https://doi.org/10.1016/j.checat.2021.10.005>
Research, vol. 2021, Article ID 979432, 2021, <https://doi.org/10.34133/2021/9794329>

Reply: Thank you for your valuable comment and literature recommendation. We have carefully read the two literature and the papers deepen our understanding of photothermocatalysis. Some metals such as Au, Ag and Cu possess specific light absorption for plasmon resonance. Such absorption is relevant with the size of metal. The plasmon-resonance absorption of metallic In has not been reported. Here, the

light-to-heat conversion of In-Em In_2O_3 is mainly realized by metallic In as shown in Supplementary Fig. 17. However, it is not certain which of the light absorption modes from metallic In exhibit the highest efficiency of light-to-heat conversion, including interband-transition absorption, intraband-transition absorption, plasmon-resonance absorption.

We would like to supplement the related discussion in the last paragraph of the “TOF activity over In-Em In_2O_3 is 866 times higher than that over In_2O_3 under light irradiation” part as below:

“Photothermal conversion over the system is one of the key factors dictating the photothermocatalytic performance of CO_2 reduction. The photothermal conversion capability of In_2O_3 is rather low (Supplementary Fig. 17a) because of the light absorption of the wavelength below 500 nm and dominant radiative emission. Compared with In_2O_3 , In-Em In_2O_3 displayed a more efficient photothermal conversion due to full-spectral light absorption (to near-infrared light) and a high probability of nonradiative relaxation (Supplementary Fig. 17). The photothermal effect of In-Em In_2O_3 originates from oxygen defects and light absorption characteristics of metallic In. As reported, oxygen defects can create mid-gap energy state and thus increase light-to-heat conversion due to enhanced light absorption⁷ and “trap-assisted recombination”⁴⁶. Compared with oxygen defects, metallic In of In-Em In_2O_3 plays a dominant role in light-to-heat conversion which heats up the metal lattice by electron-phonon scattering. Due to superior thermal conduction of metallic In, the concentrated energy in metallic In is then rapidly transferred to the active site of In_2O_3 portion for CO_2 reduction via phonon-phonon relaxation⁴⁷. However, in the future, it is worth investigating which of the light absorption modes from metallic In exhibit the highest efficiency of light-to-heat conversion, including interband-transition absorption, intraband-transition absorption, plasmon-resonance absorption⁴⁶⁻⁴⁸.”

3. In Figure 5a, the reactant and intermediate adsorption energies in In- In_2O_3 are lower than those in In_2O_3 . But it is not clear what are those adsorbate binding configurations on surfaces. The authors should provide these structures in figures in the supporting information.

Reply: Thank you for your valuable comment. We have supplemented the related figures and made a discussion. The figures are in Supplementary Fig. 39 and 40. The relevant discussions were added in page 45 and 46 of Supplementary information.

Fig. R15 The adsorbate binding configurations on the surface of In₂O₃. (a) Adsorbed CO₂ and H. (b) COOH intermediate. (c) CO and OH from COOH dissociation.

“The C atom and one of the O atoms of CO₂ are bound with O and In of In₂O₃ around oxygen vacancy, respectively. The two C-O bonds of CO₂ form a bond angle of 120°, suggesting CO₂ activation. The bent structure of CO₂ is the feature of CO₂ anion¹⁶. With the collision coupling of H and the other O atom of CO₂, COOH forms accompanied by the O end linking with In atom. After the dissociation of COOH, the OH is bridged with two In atoms.”

Fig. R16 The adsorbate binding configurations on the surface of In-In₂O₃. (a) Adsorbed CO₂ and H. (b) COOH intermediate. (c) CO and OH from COOH dissociation.

“The adsorption of H and CO₂ is on the In₂O₃ side and the interaction mode is similar with the result above on In₂O₃. H and CO₂ collide at the interface to form COOH. One of the O atoms of COOH is connected with In⁰ atom. Finally, COOH is dissociated accompanied by the OH bridged with two In atoms.”

4. For In-In₂O₃, did the authors compare the adsorption energies of adsorbate (e.g., CO₂ and H) on the In and In₂O₃ portions of the surface structure, respectively? The authors need to rationalize or clarify if the catalytic active sites are on the In, In₂O₃, or both surfaces for the In-In₂O₃ catalyst.

Reply: Thank you for your suggestion. We have theoretically calculated the adsorption energies of adsorbates (e.g., CO₂ and H) on metallic In and In₂O₃, respectively. The comparison of adsorption energies is in the table as following:

Catalyst	In ₂ O ₃	Metallic In	In-In ₂ O ₃
CO ₂	0.287 eV	-0.081 eV	-0.750 eV
H	0.235 eV	-1.058 eV	-3.292 eV

Through DFT calculations, the adsorption energies of CO₂ and H on metallic In are lower than that on In₂O₃, but the adsorption energies on In-In₂O₃ (namely at the interface between metallic In and In₂O₃) are much lower compared with metallic In and In₂O₃. This in theory indicates that the active sites are neither on the metallic In nor on the In₂O₃, but most likely at the interface between metallic In and In₂O₃. Moreover, we have employed the sole metallic In as the photothermal catalyst for CO₂ reduction, which showed that the metallic In does not have the capability for CO₂ reduction. This experimentally excludes metallic In as the active sites. Accordingly, the active sites are not the In₂O₃ and metallic In but rather the oxygen defects at their interface.

We would like to supplement the related discussion in the “Enhanced CO₂ adsorption and activation” part as below:

“The adsorption energies of CO₂ and H on metallic In were calculated to be -0.081 and -1.058 eV, respectively. The adsorption energies of CO₂ and H on In-In₂O₃ (namely at the interface between metallic In and In₂O₃) are much lower than that on In₂O₃ and metallic In, which most likely evidences that the active sites are at the interface between metallic In and In₂O₃. This is consistent with the conclusion above that the oxygen defects at the interface function as the active sites for CO₂ reduction.”

Other revisions for editors:

1. Correct the note of “see supplementary information”.
2. We have read the formatting instructions carefully and revised some incorrect formats, e.g. the word number in the abstract (less than 150), the panel style in the figures and the placement position of ACKNOWLEDGEMENTS, AUTHOR CONTRIBUTIONS and COMPETING INTERESTS.
3. We have added a “Data Availability” section after the Methods section but before the References.

That’s all. Thank you again for discussing about our study.

Yours sincerely,

OUYANG Shuxin, PhD, Professor

Key Laboratory of Pesticide and Chemical Biology of Ministry of Education

College of Chemistry,

Central China Normal University,

Wuhan 430079, China

E-mail: oysx@mail.ccnu.edu.cn